# SCALABLE CONTINUOUS-TIME HIDDEN MARKOV MODELS

## ABSTRACT

As a probabilistic tool for irregularly-sampled data, the Continuous-Time Hidden Markov Model (CTHMM) inherently handles real phenomena with uncertainties modelled by distributions. However, CTHMM is affected by (i) the costly matrix exponentiation (cubic time complexity w.r.t. the number of hidden states) involved in the estimation of transition probabilities, and (ii) the use of simplistic parametric observation models (*e.g.*, Gaussian). Thus, we propose scalable algorithms for CTHMM on traditional problems (learning, evaluation, decoding) to ensure tractability. Firstly, we factorise states of CTHMM into multiple binary states (*e.g.*, several $2 \times 2$ sub-problems) leading to a distributed closed-form exponentiation. We also accelerate matrix-vector products, reducing the complexity from quadratic to linearithmic. Secondly, the simplistic parametric distributions are replaced by the normalising flows (that transform simple distributions into complex data-driven distributions), accelerated by sharing few invertible neural networks among groups of hidden states. Training our approach takes few hours on a GPU, while standard CTHMMs with mere 10 hidden states take few weeks. On the largest dataset, our method scales favourably (up to $1024\times$ larger hidden states than naive CTHMM and outperforms it by 4.1 in log-likelihood). We also outperform competing HMMs with advanced solvers on downstream tasks.

## 1 INTRODUCTION

The size of modern datasets is both a blessing and a curse. While more information than ever before is available to build advanced statistical models, large-scale datasets expand computational footprint rendering many standard algorithms intractable. Moreover, for phenomena that evolve over time, it is common for observations to be collected at irregular time intervals which typically makes discrete-time methods inapplicable. For instance, in tracking the movement of wildlife across large geographical areas, locations from individual tagged animals are often observed over long periods of time. Observations can be irregular/sparse due to the limited capacity of sensor batteries, limited communication infrastructure across large geographical areas or complex environment. The data distributions are also difficult to parameterise.

The above issues prevent employment of the traditional statistical methods such as Hidden Markov Models (HMMs) (Rabiner & Juang, 1986) and their discrete-time variants. Specifically, discrete-time methods require data to be time-discretised which leads to the loss of information. Traditionally, HMMs have also used simple parametric observation models which can limit their ability in modelling more complex phenomena (Lorek et al., 2022). In contrast, CTHMM can handle observations received in irregular intervals, showing great potential in many fields such as health (Ge et al., 2024; Liu et al., 2015; 2013), finance (Nystrup et al., 2015), and information technology (Okamura et al., 2017; Wei et al., 2002). Compare to other continuous-time models (Chen et al., 2023; Fonseca et al., 2023; Zhang et al., 2023; Qin et al., 2024; Tang et al., 2022; Moreno-Pino et al., 2024), CTHMM can handle real phenomena with uncertainties modelled by distributions, and the hidden states provide insights into the underlying physical system.

Nevertheless, there are two main obstacles that limit applications of CTHMM in practice. Firstly, CTHMM suffer from a prohibitive time-complexity, which is cubic w.r.t. number of hidden states during learning, evaluation and decoding. CTHMM also suffers from numerical instability (Moler & Van Loan, 2003) when performing matrix exponential calculations. Secondly, existing CTHMMs

model the observations following standard parametric distributions, *e.g.*, Gaussians, mixtures of Gaussians or other families of simple parametric distributions (Jackson, 2011). Such a setting results in a limited capability in modelling complex observations that do not follow the above-mentioned distributions, thus limiting the applicability of CTHMMs on large and complex datasets.

In this paper, we introduce a fast state factorisation and normalising flows into CTHMM, significantly reducing their computational complexity and enhancing the modelling power. The proposed method factorises a large state space into independent binary state spaces to accelerate and stabilise matrix exponential calculations. Conditional normalising flows (Dinh et al., 2017) parameterised through neural networks help us map Gaussian priors to unknown complex data distributions. Using the proposed method, inference problems with CTHMM are simplified to the linearithmic time complexity w.r.t. the number of hidden states. The conditional normalising flows are made highly efficient through binding the conditions (*i.e.*, share some parameters among different conditions to reduce the number of parameters and speed up calculations) and calculating in parallel the emission probabilities.

In summary, our contributions are three-fold:

  i. We propose scalable forward and decoding algorithms for CTHMMs by factorising the latent Markov process into smaller, parallelised sub-problems, avoiding instability from large matrix exponentiations.

  ii. We enhance CTHMMs with conditional normalising flows to flexibly model complex observation distributions while reducing parameters through shared invertible neural networks.

  iii. We demonstrate state-of-the-art scalability and interpretability on datasets with millions of observations and up to $2^{12}$ hidden states, showing clear advantages of continuous-time over discrete-time modelling.

Notably, our method enables practical CTHMM inference within hours, supporting applications at unprecedented scale and stability—processing up to $1024\times$ more hidden states than standard CTHMMs under the same runtime budget.

**Notations** Uppercase letters indexed by time (*e.g.*, $H(t)$, $\mathbf{O}(t)$) denote random variables/vectors of a stochastic process; lowercase denote their realisations (*e.g.*, $\mathbf{o}_{t_k}$). Where realisations occur at multiple times, we define $\mathbf{o} = [\mathbf{o}_{t_1}^\top, \ldots, \mathbf{o}_{t_n}^\top]^\top$ with $t_1 < \cdots < t_n \in [0, \infty)$. Probability (conditional) densities/masses are $p(\cdot)$ ($p(\cdot|\cdot)$). Bold uppercase letters (*e.g.*, $\mathbf{P}$) denote matrices, entries $p_{i,j}$; time-indexed matrices (*e.g.*, $\mathbf{P}(t)$) have entries vary with $t$. A transition rate matrix $\mathbf{Q} \in \mathbb{Q}^{m \times m}$ satisfies $\sum_j q_{ij} = 0$, $q_{ij} \geq 0$, $\forall i \neq j$. The function $b(\cdot)$ maps binary vectors to integers (*e.g.*, $b([1, 1, 0]^\top) = 6$), with inverse $b^{-1}(\cdot)$. Symbols $\odot$ and $\otimes$ denote element-wise and Kronecker products, respectively.

## 2 METHODOLOGY

We factorise the state space into multiple binary state spaces, each of which has an independent Markov process that evolves according to its own matrix of transition rates (*e.g.*, $2 \times 2$ matrix). This substantially reduces the computational burden of the matrix exponentiation, as for example only $2 \times 2$ matrices are involved in computations, and these can be explicitly and precisely calculated with a mere $\mathcal{O}(m)$ time complexity. In contrast, the corresponding CTHMM with $2^m$ states that evolves according to a single matrix, $\mathbf{Q}$, suffers from a prohibitive $2^{\mathcal{O}(m)}$ time complexity (Moler & Van Loan, 2003) and numerical instability. However, the drawback of using many smaller transition matrices is a more restricted latent space. We counter this issue by introducing conditional normalising flows to enhance the expressive power of the model, while adding little computational complexity thanks to the sparse structure of the normalising flows. A diagram providing an overview of our approach is presented in Figure 1. We will introduce the fast learning/evaluation by the forward algorithm for our specification of CTHMM. The fast decoding (Viterbi) algorithm is introduced in Appendix A.2.

### 2.1 MODEL SPECIFICATION

We reformulate the ordinary Continuous-Time Markov Chain (CTMC) as multiple, parallel, independent, latent Markov processes denoted $\{H_1(t) : t \geq 0\}, \ldots, \{H_m(t) : t \geq 0\}$. Each

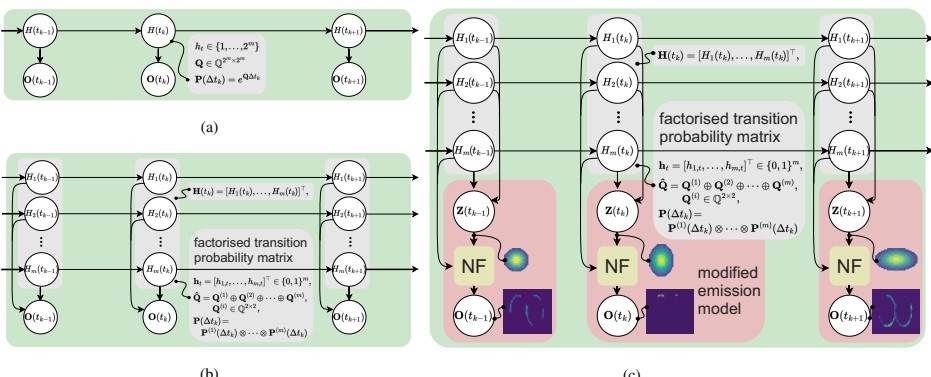

Figure 1: Markov processes evolving continuously in time (observations do not occur at regular time intervals). Fig. 1a: The standard CTHMM uses a $2^m \times 2^m$ transition rate matrix $\mathbf{Q}$. Fig. 1b: We factorise $\mathbf{Q}$ into $\mathbf{Q}^{(1)}, \ldots, \mathbf{Q}^{(m)}$ matrices of $2 \times 2$ size. This model suffers form an insufficient expressive power. Fig. 1c: Our Scalable CTHMM includes the efficient factorisation of matrix $\mathbf{Q}$ and a modified emission model with Normalising Flows (NF). The process emits a Gaussian random vector $\mathbf{Z}(t)$, conditioned on the state of each of the $m$ latent processes. $\mathbf{Z}(t)$ is transformed into a random vector, $\mathbf{O}(t)$, by the conditional normalising flows that provide the probability distribution of our observation at time $t$. The conditional normalising flows, conditioned on the state of $H_1(t), H_2(t), \ldots, H_m(t)$, transform the random variables that follow simple Gaussian distributions into more complex data distributions.

of these Markov processes is a CTMC with a binary state space indexed as $\mathcal{S}_1, \ldots, \mathcal{S}_m$. Let $\mathbf{H}(t) = [H_1(t), \ldots, H_m(t)]^\top$ and its realisation $\mathbf{h}_t = [h_{1,t}, \ldots, h_{m,t}]^\top \in \{0,1\}^m$. The hidden state variables, $h_{1,t}, \ldots, h_{m,t}$, at one timestep are mutually independent, but become conditionally dependent given the observation sequence. The likelihood of observations can be written as:

$$p(\mathbf{o}) = \sum_{\mathbf{h}_{t_1} \in \mathcal{S}_1 \times \cdots \times \mathcal{S}_m} \cdots \sum_{\mathbf{h}_{t_n} \in \mathcal{S}_1 \times \cdots \times \mathcal{S}_m} \prod_{k=1}^{n} \prod_{s=1}^{m} p(h_{s,t_k}|h_{s,t_{k-1}}) \, p(\mathbf{o}_{t_k}|\mathbf{h}_{t_k}), \tag{1}$$

where we define $p(h_{s,t_1}|h_{s,t_0}) = p(h_{s,t_1})$.

This is equivalent to assuming $p(h_{t_k}|h_{t_{k-1}}) = \prod_{s=1}^{m} p(h_{s,t_k}|h_{s,t_{k-1}})$ in ordinary CTMC. The main advantage of the parallel independent structure is faster calculation of $\prod_{s=1}^{m} p(h_{s,t_k}|h_{s,t_{k-1}})$, where the matrix exponentiation can be numerically stabilised and simplified.

**Theorem 2.1** *The $m$ parallel and independent CTMC, denoted $\{H_1(t) : t \geq 0\}, \ldots, \{H_m(t) : t \geq 0\}$, can be ensembled into a single CTMC $\{H(t) : t \geq 0\}$ whose state space is a combinatorial of component CTMCs with the corresponding transition rate matrix $\hat{\mathbf{Q}}$ given as:*

$$\hat{\mathbf{Q}} = \mathbf{Q}^{(1)} \oplus \mathbf{Q}^{(2)} \oplus \cdots \oplus \mathbf{Q}^{(m)}, \tag{2}$$

*where $\mathbf{Q}^{(1)}, \ldots, \mathbf{Q}^{(m)}$ are the transition rate matrices for CTMCs $\{H_1(t) : t \geq 0\}, \ldots, \{H_m(t) : t \geq 0\}$. Operator $\oplus$ denotes the Kronecker sum defined as $\mathbf{A} \oplus \mathbf{B} \triangleq \mathbf{A} \otimes \mathbf{I}_b + \mathbf{B} \otimes \mathbf{I}_a$ where $\mathbf{I}_a$ and $\mathbf{I}_b$ are the identity matrices of the same size as $\mathbf{A}$ and $\mathbf{B}$.*

The proof of Theorem 2.1 can be easily derived by the property of Kronecker sum: $e^{\mathbf{A}} \otimes e^{\mathbf{B}} = e^{\mathbf{A} \oplus \mathbf{B}}$. It indicates that the ensemble of smaller CTMCs, say size $2 \times 2$ (matrices of other size can also be used, *e.g.*, $3 \times 3, 4 \times 4, \ldots$), generates a CTMC with a larger but constrained state space. Conversely, not all CTMCs with a large state space can be decomposed into an ensemble of smaller CTMCs. This means that the ensembled CTMC generally will not be as expressive as an unconstrained CTMC with the same size state space.

To alleviate such a constrain, our emission distributions incorporates the normalising flows (Papamakarios et al., 2021) in contrast to the existing CTHMMs which are restricted to Gaussian/other simplistic parametric emission distributions. The observation $\mathbf{O}(t)$ is transformed to another latent random vector $\mathbf{Z}(t)$ through a composition of $l$ bijections $\boldsymbol{f}_j = f_{1,j} \circ f_{2,j} \circ \cdots \circ f_{l,j}$, conditioned on the hidden states. Thus, CTHMMs can model more complicated distributions of observations, and

also counter the limitation on transition rate matrix $\hat{\mathbf{Q}}$. The log-emission probability is calculated by a change of variables: $\log p(\mathbf{o}_t | b(\mathbf{H}(t)) = j) = \log \phi(\mathbf{z}_t | \boldsymbol{\mu}_j, \boldsymbol{\Sigma}_j) + \sum_{i=1}^{l} \log \left| \det \frac{\partial f_{i,j}}{\partial \mathbf{r}_t^{(i-1)}} \right|$, where $\mathbf{r}_t^{(0)} = \mathbf{o}_t$, $f_{i,j}(\mathbf{r}_t^{(i-1)}) = \mathbf{r}_t^{(i)}$, and $\mathbf{r}_t^{(l)} = \mathbf{z}_t$. Moreover, $\phi(\mathbf{z}_t | \boldsymbol{\mu}_j, \boldsymbol{\Sigma}_j)$ is a Gaussian probability density function with mean $\boldsymbol{\mu}_j$ and covariance matrix $\boldsymbol{\Sigma}_j$.

## 2.2 Fast Learning/Evaluation by the Forward Algorithm

---

**Algorithm 1** Fast Forward Algorithm for CTHMM.

---

**Input:** Observation $\mathbf{o} = [\mathbf{o}_{t_1}^\top, \ldots, \mathbf{o}_{t_n}^\top]^\top$; parameters $\boldsymbol{\pi}, \{\mathbf{Q}^{(s)}\}_{s=1}^m, \{\boldsymbol{f}_j\}_{j=1}^{\bar{m}}, \{\boldsymbol{\mu}_j, \boldsymbol{\Sigma}_j\}_{j=1}^{2^m}$
**Output:** The log-likelihood $p(\mathbf{o})$.

1: **for** $k = 1$ **to** $n$ **do**
2:    **for** $j = 1$ **to** $2^m$ **do**
3:       $\beta_j(\mathbf{o}_{t_k}) = \phi(\boldsymbol{f}_j(\mathbf{o}_{t_k}) | \boldsymbol{\mu}_j, \boldsymbol{\Sigma}_j)$;
4:    **end for**
5: **end for**
6: $\boldsymbol{\alpha}(1) = \boldsymbol{\pi} \odot \boldsymbol{\beta}(\mathbf{o}_{t_1})$;
7: **for** $k = 2$ **to** $n$ **do**
8:    Calculate $\mathbf{P}^{(1)}(\Delta t_k), \ldots, \mathbf{P}^{(m)}(\Delta t_k)$ with (3);
9:    use Algorithm 2 to calculate $\hat{\boldsymbol{\alpha}}(k) = \left( \bigotimes_{s=1}^m \mathbf{P}^{(s)}(\Delta t_k) \right)^\top \boldsymbol{\alpha}(k-1)$;
10:   $\boldsymbol{\alpha}(k) = \hat{\boldsymbol{\alpha}}(k) \odot \boldsymbol{\beta}(\mathbf{o}_{t_k})$;
11: **end for**
12: $p(\mathbf{o}) = \mathbf{1}^\top \boldsymbol{\alpha}(n)$.

---

The forward algorithm, essential to learning and evaluation of CTHMM, is intractable for ordinary CTHMM on large datasets even for a very small state space (*e.g.*, size 10). With the factorial structure of CTHMM, we implement a stable and scalable forward algorithm by 1) using the closed-form expression for matrix exponentiation; 2) using an efficient method for computing the matrix-vector product involving Kronecker products; 3) reducing the number of normalising flows by grouping hidden states. The forward algorithm helps efficiently calculate the likelihood $p(\mathbf{o})$, *i.e.*, the evaluation task. One can use modern optimisation toolkits (Abadi et al., 2016; Paszke et al., 2019) providing automatic differentiation/GPU acceleration for fast learning of model parameters.

To perform the fast forward algorithm, we parameterise the CTHMM as $\{\boldsymbol{\pi}, \{\mathbf{Q}^{(s)}\}_{s=1}^m, \{\boldsymbol{f}_j\}_{j=1}^{\bar{m}}, \{\boldsymbol{\mu}_j, \boldsymbol{\Sigma}_j\}_{j=1}^{2^m}\}$. Define $\boldsymbol{\pi} = \left[ \Pr(b(\mathbf{H}(t_1)) = 1), \ldots, \Pr(b(\mathbf{H}(t_1)) = 2^m) \right]^\top$ and $\mathbf{Q}^{(s)} = \left[ -q_0^{(s)}, q_0^{(s)}; q_1^{(s)}, -q_1^{(s)} \right]$, where $q_0^{(s)}, q_1^{(s)} > 0$. The set $\{\mathbf{Q}^{(s)}\}_{s=1}^m$ contains the transition rate matrices for each of the independent Markov processes. Sets $\{\boldsymbol{f}_j\}_{j=1}^{\bar{m}}$ and $\{\boldsymbol{\mu}_j, \boldsymbol{\Sigma}_j\}_{j=1}^{2^m}$ contain parameters for the emission model (*i.e.*, $\bar{m}$ invertible neural networks and $2^m$ means and covariances for Gaussian distributions of the normalising flows), where $0 \le \bar{m} \le 2^m$ is the number of normalising flows.

---

**Algorithm 2** Fast Calculate $\hat{\boldsymbol{\alpha}} = \left( \bigotimes_{s=1}^m \mathbf{P}^{(s)} \right)^\top \boldsymbol{\alpha}$.

---

**Input:** $\mathbf{P}^{(1)}, \ldots, \mathbf{P}^{(m)} \in \mathbb{R}^{2 \times 2}$, $\boldsymbol{\alpha} \in \mathbb{R}^{2^m}$;

**Output:** $\hat{\boldsymbol{\alpha}} = \left( \bigotimes_{s=1}^m \mathbf{P}^{(s)} \right)^\top \boldsymbol{\alpha}$.

1: **for** $s = 1$ **to** $m$ **do**
2:    $\boldsymbol{\alpha} = \text{reshape}(\boldsymbol{\alpha}, 2, 2^{m-1})$;
3:    $\boldsymbol{\alpha} = \boldsymbol{\alpha}^\top \mathbf{P}^{(s)}$;
4: **end for**
5: $\hat{\boldsymbol{\alpha}} = \text{flatten}(\boldsymbol{\alpha})$.

---

For a sequence of $n$ observations $\mathbf{o} = [\mathbf{o}_{t_1}^\top, \ldots, \mathbf{o}_{t_n}^\top]^\top$, we recursively compute the following quantities:

$$\alpha_j(k) = p(\mathbf{o}_{t_1}, \ldots, \mathbf{o}_{t_k}, b(\mathbf{H}(t_k)) = j),$$
$$\boldsymbol{\alpha}(k) = [\alpha_1(k), \ldots, \alpha_{2^m}(k)]^\top,$$
$$\beta_j(\mathbf{o}_{t_k}) = p(\mathbf{o}_{t_k} | b(\mathbf{H}(t_k)) = j),$$
$$\boldsymbol{\beta}(\mathbf{o}_{t_k}) = [\beta_1(\mathbf{o}_{t_k}), \ldots, \beta_{2^m}(\mathbf{o}_{t_k})]^\top,$$
$$\Delta t_k = t_k - t_{k-1}, (k > 1),$$
$$\mathbf{P}^{(s)}(\Delta t_k) = e^{\mathbf{Q}^{(s)} \Delta t_k},$$

where we use $\beta_j(\cdot)$ as a concise representation of the conditional probability distribution of $\mathbf{O}(t)$ given $b(\mathbf{H}(t_k)) = j$.

The forward Algorithm 1 calculates the above quantities recursively until one obtains the vector $\boldsymbol{\alpha}(n)$ at last observation, summing over which gives $p(\mathbf{o})$. A bottleneck for the forward algorithm is calculating transition probability matrices $\mathbf{P}(\Delta t_k) = e^{\mathbf{Q}\Delta t_k}$ (where $\Delta t_k = t_k - t_{k-1}$) by the matrix exponentiation, followed by the product with $\boldsymbol{\alpha}(k-1)$ and $\boldsymbol{\beta}(\mathbf{o}_{t_k})$ to obtain $\boldsymbol{\alpha}(k)$. Thanks to our model specification, we only need to perform matrix exponentiation for a collection of $m$ small matrices, $\mathbf{Q}^{(1)}, \ldots, \mathbf{Q}^{(m)}$, each of size $2 \times 2$. The matrix-vector product also benefits from this specific factorising structure. These steps correspond to lines 8-10 in Algorithm 1. In contrast, a non-factorised naive approach requires performing exponentiation of matrix of size $M \times M$ (where $M = 2^m$) which requires an Eigenvalue Decomposition (EigD) or Singular Value Decomposition

(SVD), which generally have complexity $\mathcal{O}(M^3)$. Evaluating exponentiation $n-1$ times for such a large matrix is intractable. Moreover, EigD and SVD suffer from instability (Moler & Van Loan, 2003), also during backpropagation which is undefined if the non-simple eigenvalues/singular values occur (*i.e.*, $\lambda_i = \lambda_j : i \neq j$) (Koniusz & Zhang, 2022).

Figure 2: The $\bar{m} = 2$ conditional normalising flows for 4 hidden states. The branch in the blue colour (solid lines) shows that an observation $\mathbf{o}_t$ is processed by the bijective function $\boldsymbol{f}_2$. The transformed variable $\mathbf{z}_t$ is evaluated with the Gaussian probability density function $\phi(\mathbf{z}_t|\boldsymbol{\mu}_4, \boldsymbol{\Sigma}_4)$. The probability densities $\phi(\mathbf{z}_t|\boldsymbol{\mu}_4, \boldsymbol{\Sigma}_4)$ are then transformed back to the density $\beta_4(\mathbf{o}_t) = p(\mathbf{o}_t|b(\mathbf{H}(t)) = 4)$ by a change of variables. The yellow box indicates that a group of hidden states shares an invertible neural net $\boldsymbol{f}_2$.

Given the transition rate matrix $\mathbf{Q}^{(s)}$, the matrix $\mathbf{P}^{(s)}(\Delta t_k)$ enjoys a closed form:

$$\mathbf{P}^{(s)}(\Delta t_k) = \frac{1}{q_\Sigma^{(s)}} \begin{bmatrix} q_1^{(s)} + q_0^{(s)} e^{-q_\Sigma^{(s)} \Delta t_k} & q_0^{(s)} - q_0^{(s)} e^{-q_\Sigma^{(s)} \Delta t_k} \\ q_1^{(s)} - q_1^{(s)} e^{-q_\Sigma^{(s)} \Delta t_k} & q_0^{(s)} + q_1^{(s)} e^{-q_\Sigma^{(s)} \Delta t_k} \end{bmatrix}, \quad (3)$$

where $q_\Sigma^{(s)} = q_0^{(s)} + q_1^{(s)}$. The element in the $a$-th row and $b$-th column represents $\Pr(H_s(t_k) = b|H_s(t_{k-1}) = a)$. The full transition matrix considering all the states becomes:

$$\mathbf{P}(\Delta t_k) = \mathbf{P}^{(1)}(\Delta t_k) \otimes \cdots \otimes \mathbf{P}^{(m)}(\Delta t_k), \quad (4)$$

with elements $p_{i,j}(\Delta t_k) = \Pr(b(\mathbf{H}(t_k)) = j|b(\mathbf{H}(t_{k-1})) = i)$. Thus, the calculation of the full transition matrix $\mathbf{P}(\Delta t_k)$ is split into smaller tasks of exponentiation of matrices of $2 \times 2$ size. Note that in practice, we neither calculate nor store $\mathbf{P}(\Delta t_k)$ explicitly. Instead we compute its product with the vector $\boldsymbol{\alpha}(k-1)$ directly without expanding to $\mathbf{P}(\Delta t_k)$ using Algorithm 2. This reduces the time and space complexity even further. The fast computation of products of Kronecker products of matrices with a vector is known as the *shuffle algorithm* (Fernandes et al., 1998; De Boor, 1979; Pereyra & Scherer, 1973). More details on the shuffle algorithm can be found in the Appendix A.1. However, it does not seem to be widely recognised that no shuffling is actually needed (Fackler, 2019). We implement Algorithm 2 with only matrix-matrix multiplication and matrix reshape operations. This fast subroutine helps evaluate products of Kronecker products of matrices with a vector, *i.e.*, $(\bigotimes_{s=1}^{m} \mathbf{P}^{(s)})^\top \boldsymbol{\alpha}$ by saving both time and memory. In contrast to the matrix exponential that is applied to a large transition rate matrix, $\mathbf{Q} \in \mathbb{Q}^{2^m \times 2^m}$, followed by a product with a length-$2^m$ vector, the procedures in Algorithm 1 are significantly accelerated and stabilised.

Another challenge in the forward algorithm is calculating the emission probabilities $\beta_j(\mathbf{o}_{t_k}) = p(\mathbf{o}_{t_k}|b(\mathbf{H}(t_k)) = j), \forall j \in \{1, \ldots, 2^m\}$, corresponding to lines 1-5 in Algorithm 1. There needs to be $2^m$ bijections conditioned on $\mathbf{H}(t_k)$, and these bijective functions are often implemented through invertible neural networks, which may introduce prohibitive space and time complexity. Thus, we group the hidden states that emission probabilities are conditioned on into $\bar{m}$ equal-size groups, leading to $\boldsymbol{f}_j \triangleq \boldsymbol{f}_{((j-1) \bmod \bar{m})+1}$, where $\bmod$ is the modulo operator. As a result, we only need $\bar{m}$ invertible neural networks, as illustrated in Figure 2. Note that one may be able to find better bindings of hidden states, but practically, the simple mapping $((j-1) \bmod \bar{m}) + 1$ is sufficient. The limited count of the normalising flows is thus reflected in the fact that bijections only change between groups of hidden states rather than for each hidden state, which can be viewed as a parameter-sharing scheme, where the hidden states within each group share the same bijections.

Figure 2 shows that an observation $\mathbf{o}_t$ is processed by the bijective functions (*e.g.*, in parallel), and the transformed variables $\mathbf{z}_t$ are evaluated with corresponding Gaussian probability density functions. The probabilities from the Gaussian probability density functions are then transformed back to the probabilities of $\mathbf{o}_t$ conditioned on different $\mathbf{H}(t)$ by a change of variables, obtaining $\beta_j(\mathbf{o}_t) = p(\mathbf{o}_t|b(\mathbf{H}(t)) = j)$. Sharing invertible neural networks by the conditional normalising flows, combined with Algorithms 1 and 2 make the forward algorithm highly efficient, accelerating learning of model parameters.

**Prediction.** For prediction, we need to obtain the probability of $\mathbf{o}_{t_k}$ given the elapsed time $\Delta t_k = t_k - t_{k-1}$ and all previous observations $\mathbf{o}_{t_1}, \ldots, \mathbf{o}_{t_{k-1}}$, that is $p(\mathbf{o}_{t_k}|\mathbf{o}_{t_1}, \ldots, \mathbf{o}_{t_{k-1}}) = \frac{p(\mathbf{o}_{t_1}, \ldots, \mathbf{o}_{t_k})}{p(\mathbf{o}_{t_1}, \ldots, \mathbf{o}_{t_{k-1}})} = \frac{\hat{\boldsymbol{\alpha}}(k)^\top \boldsymbol{\beta}(\mathbf{o}_{t_k})}{p(\mathbf{o}_{t_1}, \ldots, \mathbf{o}_{t_{k-1}})}$. Notice that the probability is modelled by a mixture of $2^m$ compo-

nents, with the $j$-th component corresponding to the density function $\beta_j(\mathbf{o}_{t_k}) = p(\mathbf{o}_{t_k}|b(\mathbf{H}(t_k)) = j)$ and weighted by normalising $\hat{\boldsymbol{\alpha}}(k)$ to have the unit sum. Thus, the prediction step can be easily performed with a slight modification of Algorithm 1, rapidly calculating $\hat{\boldsymbol{\alpha}}(k)$ and $\beta_j(\mathbf{o}_{t_k})$.

**Complexity Analysis.** The complexity of conditional normalising flows depends on the choice of bijective neural networks. We adopt a simple architecture with $\bar{m} \leq 8$ invertible networks, computed in parallel for efficiency. With $m$ independent Markov processes and $n$ observations, Algorithm 1 has time $\mathcal{O}(nM \log(M))$ with $M = 2^m$ hidden states. This is substantially lower than the ordinary CTHMM, where matrix exponentiation of the $M \times M$ transition matrix has $\mathcal{O}(nM^3)$ complexity. Our specification also reduces the matrix-vector product from $\mathcal{O}(M^2)$ to $\mathcal{O}(M \log(M))$ in Algorithm 2. The gap widens when repeatedly running the forward algorithm, as required for learning and evaluation. Similar reductions hold for decoding and space complexity.

## 3 RELATED WORKS

Continuous-time models are more flexible than discrete-time models in principle because they can handle data irregularity in time. They have shown promising performance in various applications (Chen et al., 2023; Fonseca et al., 2023; Zhang et al., 2023; Qin et al., 2024; Tang et al., 2022; Moreno-Pino et al., 2024). Among them, CTHMM excels in probabilistic modelling, outputting distributions rather than point estimation. The physical meaning of hidden states helps understand the underlying system mechanism. Prior works have shown that the matrix exponentiation in the CTHMM is notoriously slow, unstable and not scalable (Moler & Van Loan, 2003). Thus, application of CTHMMs are hindered despite being a natural choice for modelling continuous-time data. To address such a shortcoming, Liu et al. (2015) proposed a few efficient learning algorithms for CTHMMs, but the efficiency came with a prerequisite that a dataset should have a small number of distinct time intervals. Other learning algorithms (Jackson, 2011; Dempsey et al., 2017; Leiva-Murillo et al., 2011; Nodelman et al., 2005; Spaeh & Tsourakakis, 2024) were proposed but they do not address the problem of scalability. We improve the scalability of the CTHMM by using a specific model structure inspired by Factorial HMMs (Ghahramani & Jordan, 1997). Extension of the Factorial HMM to the CTHMM is elegant, helping us develop a fast learning algorithm analogous to that of Schweiger et al. (2019). The fast computation with Kronecker products was inspired by Fackler (2019). We expand the number of hidden states from few hundreds at most (Liu et al., 2015) to few thousands, while being still able to further promote the expressive power with the efficient normalising flows.

Existing CTHMMs typically assume emission models with a relatively simple parametric distributions, *e.g.*, Gaussian distribution (Liu et al., 2015), some discrete distributions (Wei et al., 2002; Bureau et al., 2003), or using generalised linear models (Dempsey et al., 2017). This limits their ability to model data following complex and unknown probability distributions that can be learned from large datasets. Unfortunately, even for discrete-time HMMs, there is an absence of use cases with more advanced emission models. Lorek et al. (2022) has proposed to use normalising flows (Papamakarios et al., 2021) for discrete-time HMM. This method is however limited to very small state spaces ($2 \sim 4$ hidden states) due to the high complexity in discretising observations for parameter learning and normalising flows. Normalising flows have shown great potential in various applications (Wong et al., 2020; Kanwar et al., 2020; Lugmayr et al., 2020) but their complexity grows with the number of hidden states, making it unsuitable in applications with large state spaces. The complexity in continuous-time settings makes this even more challenging.

## 4 EXPERIMENTS

We conduct several experiments to evaluate and substantiate these claims: i. our method efficiently and effectively scales up to large state space/datasets (Table 2, Fig. 4); ii. models with large state space perform better on complicated large datasets (Table 2, Fig. 4); iii. models with large state space perform better in downstream tasks (Table 3, Fig. 5); iv. continuous-time models perform better on irregularly observed data (Fig. 6); v. normalising flows contribute to the performance improvement (Fig. 6). The experiments are designed to show the scalability of our method. They also show the benefits and necessity of using the large state space and normalising flows in addition to the advantages of continuous-time modelling.

### 4.1 DATASETS

Table 1: Statistics of the datasets in our experiments. (lens:lengths; seqs:sequences; $\mu$:mean; $\sigma$:standard deviation)

| Dataset | lens of seqs | #seqs | $\Delta t/s\,(\mu \pm \sigma)$ | #observations |
|---|---|---|---|---|
| Taxi | $3 \sim 11,871$ | 3,127 | $415.57 \pm 2,173.74$ | 3,845,410 |
| RAATD | $3 \sim 17,067$ | 4,009 | $11,406.10 \pm 41,409.11$ | 2,570,183 |
| LRFF | $11 \sim 1,853$ | 51 | $24,429.81 \pm 67,085.02$ | 20,300 |

We use three datasets with complex stochastic dynamics that contain two-dimensional geographic locations data over extensive spatial areas for multiple tracked objects, each yielding its own sequences or trajectories. We visualise areas in the two large-scale datasets containing most of observations in Fig. 3. More details and a complete view of the two large-scale datasets can be found in Appendix A.3.3. We summarise statistics of all datasets in Table 1.

### 4.2 BASELINES, IMPLEMENTATION DETAILS, AND EVALUATION METRICS

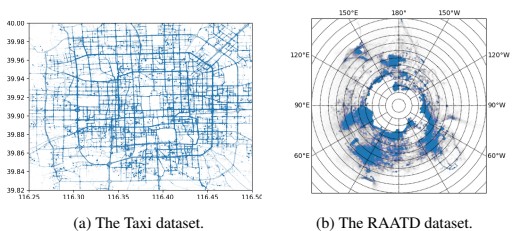

(a) The Taxi dataset.    (b) The RAATD dataset.

Figure 3: Visualisation of observations in two datasets.

We compare the Scalable CTHMM to three baselines: (i) CTHMM, the latest CTHMM from the `msm` package (Jackson, 2011; 2024); (ii) HMM, a discrete-time model with discretised time steps to approximate continuous time (Liu et al., 2015); and (iii) FaHMM, the factorial HMM (Ghahramani & Jordan, 1995) implemented with the fast algorithm of Schweiger et al. (2019). It is a discrete-time model so we also adapted it to handle continuous time data by discretisation. For discrete-time models, we discretise irregular intervals by computing mean time differences across the data, normalising, and rounding to integers. Transition probabilities are then obtained from powers of the transition probability matrix. Emissions are restricted to Gaussian densities for scalability and stability. All baselines are trained with the forward algorithm (enabling mini-batch training), except the ordinary CTHMM, which uses the solver from the `msm` package. In our model, we set the number of invertible neural networks $\bar{m} = \min(8, \#\text{hidden states})$, using 3 coupling layers (Dinh et al., 2017; 2015) with ActNorm (Kingma & Dhariwal, 2018). Training employs AdamW (Loshchilov & Hutter, 2018). Models are trained on the first 80% of each sequence to minimise average negative log-likelihood (per observation) and evaluated on each complete sequences. We found that these datasets were too complex to overfit, so we did not use validation sets. Models with $2^1 \sim 2^{12}$ hidden states are trained for 100 epochs, except the ordinary CTHMM. Experiments were run with NVIDIA Tesla P100 GPUs (16GB) or AMD EPYC 7543 CPUs (when more than 16GB memory was required), with a 160-hour time limit. We evaluate models using average negative log-likelihood (-LogLik), continuous ranked probability score (CRPS) (Matheson & Winkler, 1976), and training time. CRPS (Matheson & Winkler, 1976) is analogous to the mean square error between the predicted cumulative density function (CDF) and the true CDF. It degenerates to the mean absolute error if the predicted distribution is a degenerate distribution (*e.g.*, a point estimation). More details are in Appendix A.4.

### 4.3 EXPERIMENTAL RESULTS

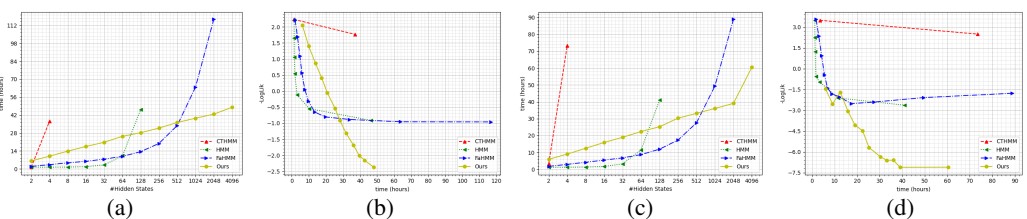

(a)      (b)      (c)      (d)

Figure 4: Time *vs.*-LogLik and #hidden states on the Taxi ((a), (b)) and RAATD ((c), (d)) datasets on negative log-likelihood minimisation. Longer running indicates a larger state space.

Table 2: Experimental results on Taxi, RAATD, and LRFF datasets within 160 hours (↓: smaller is better; M: the number of hidden states; -LogLik: negative log-likelihood; CRPS: continuous ranked probability score).

| M | Metrics↓ | Taxi Dataset | | | | RAATD Dataset | | | | LRFF Dataset | | | |
|---|---|---|---|---|---|---|---|---|---|---|---|---|---|
| | | CTHMM | HMM | FaHMM | Ours | CTHMM | HMM | FaHMM | Ours | CTHMM | HMM | FaHMM | Ours |
| 2 | -LogLik | 2.234 | 2.220 | 2.218 | 2.051 | 3.485 | 3.474 | 3.551 | -1.460 | 1.922 | 2.008 | 1.926 | 0.628 |
| | CRPS | 0.498 | 0.495 | 0.497 | 0.492 | 0.487 | 0.508 | 0.482 | 0.464 | 0.366 | 0.463 | 0.438 | 0.349 |
| | Time (s) | 4436 | 4841 | 6986 | 22240 | 12654 | 4429 | 6409 | 21451 | 50 | 15 | 25 | 89 |
| 4 | -LogLik | 1.768 | 1.659 | 1.697 | 1.400 | 2.492 | 2.227 | 2.357 | -2.535 | 1.347 | 1.383 | 1.958 | 0.079 |
| | CRPS | 0.392 | 0.416 | 0.380 | 0.390 | 0.391 | 0.351 | 0.360 | 0.338 | 0.314 | 0.379 | 0.399 | 0.279 |
| | Time (s) | 133575 | 4850 | 11868 | 35661 | 264116 | 4528 | 10662 | 32636 | 1713 | 17 | 39 | 139 |
| 8 | -LogLik | - | 1.056 | 1.094 | 0.872 | - | 1.242 | 0.927 | -1.703 | 1.274 | 1.706 | 0.738 | -1.004 |
| | CRPS | - | 0.279 | 0.283 | 0.287 | - | 0.266 | 0.250 | 0.425 | 0.279 | 0.412 | 0.341 | 0.271 |
| | Time (s) | - | 5117 | 16979 | 49752 | - | 4925 | 15204 | 45005 | 14096 | 17 | 58 | 189 |
| 16 | -LogLik | - | 0.543 | 0.565 | 0.414 | - | -0.571 | -0.431 | -3.073 | - | 0.437 | -0.341 | -1.570 |
| | CRPS | - | 0.219 | 0.218 | 0.223 | - | 0.163 | 0.174 | 0.240 | - | 0.402 | 0.339 | 0.169 |
| | Time (s) | - | 6305 | 21232 | 63003 | - | 6150 | 19729 | 57357 | - | 25 | 77 | 241 |
| 32 | -LogLik | - | -0.114 | 0.052 | -0.052 | - | -0.949 | -1.389 | -4.068 | - | 0.434 | -0.532 | -2.210 |
| | CRPS | - | 0.159 | 0.169 | 0.169 | - | 0.159 | 0.133 | 0.162 | - | 0.426 | 0.353 | 0.137 |
| | Time (s) | - | 11054 | 27359 | 74903 | - | 11802 | 24538 | 68469 | - | 58 | 101 | 286 |
| 64 | -LogLik | - | -0.547 | -0.313 | -0.544 | - | -2.114 | -1.830 | -4.477 | - | 0.168 | -0.915 | -2.680 |
| | CRPS | - | 0.127 | 0.141 | 0.132 | - | 0.114 | 0.133 | 0.139 | - | 0.421 | 0.365 | 0.130 |
| | Time (s) | - | 36916 | 35172 | 91503 | - | 41585 | 31516 | 80657 | - | 145 | 131 | 336 |
| 128 | -LogLik | - | -0.911 | -0.656 | -0.915 | - | -2.634 | -2.020 | -5.686 | - | 0.322 | -1.143 | -2.899 |
| | CRPS | - | 0.106 | 0.129 | 0.113 | - | 0.093 | 0.282 | 0.078 | - | 0.417 | 0.376 | 0.125 |
| | Time (s) | - | 166889 | 48410 | 101615 | - | 147624 | 43092 | 91004 | - | 618 | 192 | 395 |
| 256 | -LogLik | - | - | -0.805 | -1.315 | - | - | -2.523 | -6.357 | - | 0.327 | -1.385 | -3.185 |
| | CRPS | - | - | 0.124 | 0.100 | - | - | 0.097 | 0.066 | - | 0.418 | 0.417 | 0.126 |
| | Time (s) | - | - | 71456 | 114676 | - | - | 62516 | 109112 | - | 2417 | 283 | 440 |
| 512 | -LogLik | - | - | -0.883 | -1.686 | - | - | -2.409 | -6.608 | - | 0.450 | -1.661 | -3.082 |
| | CRPS | - | - | 0.127 | 0.094 | - | - | 0.115 | 0.053 | - | 0.417 | 0.389 | 0.167 |
| | Time (s) | - | - | 121856 | 130317 | - | - | 99152 | 119394 | - | 14065 | 477 | 482 |
| 1024 | -LogLik | - | - | -0.955 | -2.010 | - | - | -2.087 | -6.601 | - | - | -1.691 | -3.115 |
| | CRPS | - | - | 0.125 | 0.090 | - | - | 0.124 | 0.051 | - | - | 0.422 | 0.117 |
| | Time (s) | - | - | 229562 | 142333 | - | - | 177942 | 129805 | - | - | 872 | 544 |
| 2048 | -LogLik | - | - | -0.959 | -2.153 | - | - | -1.777 | -7.113 | - | - | -1.438 | -3.283 |
| | CRPS | - | - | 0.127 | 0.090 | - | - | 0.445 | 0.049 | - | - | 0.449 | 0.164 |
| | Time (s) | - | - | 420024 | 153645 | - | - | 319682 | 140634 | - | - | 1657 | 590 |
| 4096 | -LogLik | - | - | - | -2.374 | - | - | - | -7.103 | - | - | -1.716 | -3.139 |
| | CRPS | - | - | - | 0.093 | - | - | - | 0.077 | - | - | 0.447 | 0.114 |
| | Time (s) | - | - | - | 173070 | - | - | - | 217800 | - | - | 3300 | 647 |

Table 3: Results on 4 downstream tasks on the RAATD dataset (AUC: area under the ROC curve; ACC: accuracy).

| Task | Metrics | CTHMM | HMM | FaHMM | Ours |
|---|---|---|---|---|---|
| i | AUC | 0.578 | 0.946 | 0.963 | **0.982** |
| | ACC | 0.232 | 0.638 | 0.759 | **0.822** |
| ii | AUC | 0.623 | 0.785 | 0.858 | **0.941** |
| | ACC | 0.832 | 0.832 | 0.843 | **0.875** |
| iii | AUC | 0.530 | 0.762 | 0.707 | **0.793** |
| | ACC | 0.664 | 0.736 | 0.692 | **0.750** |
| iv | AUC | 0.641 | **0.966** | 0.877 | 0.940 |
| | ACC | 0.382 | **0.810** | 0.691 | 0.768 |

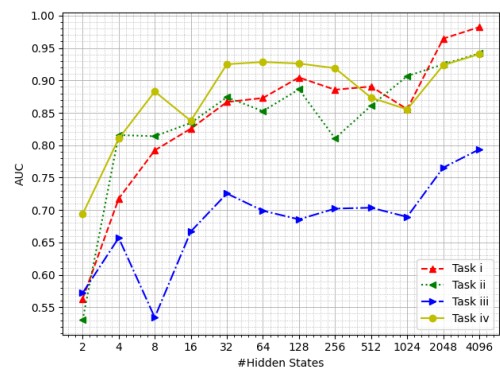

Figure 5: AUC *vs.* the number of hidden states on downstream tasks of the RAATD dataset.

**Results on Negative Log-Likelihood Minimisation.** Results in Table 2 with "-" indicate models requiring over 160 hours or 128GB memory. On the Taxi dataset, the ordinary CTHMM scales only to 4 hidden states, showing no performance advantages in this complicated dataset due to the small latent space, convergence issues, and numerical instability. HMMs are generally better than FaHMMs and our model with $32 \sim 128$ hidden states thanks to their unrestricted state space, but fail to scale further since storing full transition matrices is infeasible. Our method surpasses FaHMM thanks to continuous-time modelling and normalising flows, and unlike FaHMM, scales to 4096 states. Time discretisation limits FaHMM due to costly matrix powers on long intervals. Results on the RAATD dataset (Table 2) follow a similar pattern. Results on the LRFF dataset (Table 2) shows that our model performs best across all number of hidden states. Compared to HMM, our method takes much less time as $M$ increases. Also, HMM takes more than 128G memory with 1024

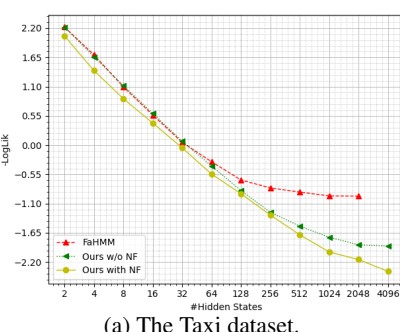

(a) The Taxi dataset.

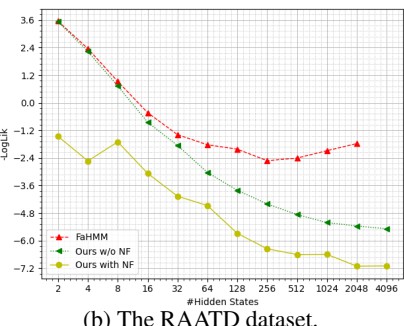

(b) The RAATD dataset.

Figure 6: Comparing models of discrete-time (FaHMM), without normalising flows (FaHMM, Ours w/o NF) and Scalable CTHMM model (Ours with NF).

hidden states, as it has to store full transition probability matrices for every time step. On the Taxi and RAATD datasets, we compare (i) hidden states *vs*. running time and (ii) running time *vs*. negative log-likelihood in Fig. 4. Longer running times indicates larger state spaces. Each marker denotes a model with predefined hidden states, trained for 100 epochs. Only our method scales to 4096 hidden states with approximately linearithmic time complexity, showing both efficiency and effectiveness within a reasonable budget. These results highlight the benefits of scalable methods for large state spaces and datasets. Predicted heatmaps sampled from our model are shown in Appendix A.5.1.

**Results on Decoding for Downstream Tasks.** We evaluate on the RAATD dataset with four downstream tasks based on decoding results. For each method, we use the trained model with the largest number of hidden states and apply Viterbi decoding to each sequence. The decoded hidden-state sequences serve as inputs for: i) species classification, ii) maturity classification, iii) sex classification, iv) breeding stage classification. More details are in Appendix A.5). Since not all observations are labelled, we only train and test on labelled data. As in the negative log-likelihood experiments, we train on 80% of hidden states per sequence and evaluate on the full sequence. For the first three time-invariant tasks, we average decoded states as input to a multi-layer perceptron (MLP). For breeding stage classification, each observation's hidden state is used directly with a similar MLP. Results in Table 3 show our method performs best on 3 of 4 tasks, and second-best on task iv. This highlights the benefits of large state spaces and the utility of hidden states for downstream tasks. For task iv, performance is likely limited by few sequences have "changing breeding stage" and the simplicity of pointwise (*vs*. sequence) models.

### 4.4 ABLATION STUDY

**Benefits of Large State Space in Downstream Tasks.** We conduct experiments on downstream tasks with our models of different number of hidden states. The results are illustrated in Fig. 5. There is a trend of increasing AUC as the number of hidden states increases.

**Benefits of Continuous-time Modelling and Normalising Flows.** To show the benefits of continuous-time modelling and normalising flows, in Fig. 6 we run experiments with our intermediate model by removing normalising flows from Scalable CTHMM (Fig. 1b). We denote such an intermediate model as "Ours w/o NF". The difference between FaHMM and "Ours w/o NF" is accredited to the continuous-time modelling. The improvement of "Ours with NF" (Fig. 1c) over "Ours w/o NF" verifies the effectiveness of using normalising flows in emission distributions.

### 5 CONCLUSION AND LIMITATIONS

We provide a highly efficient and expressive model, called Scalable CTHMM, for complex phenomena that evolve continuously in time. It incorporates normalising flows with a scalable forward algorithm. The proposed model makes the CTHMM practical and applicable on large scale datasets, removing many of the computational limitations of ordinary CTHMMs. However, the use of factorisation in Scalable CTHMM may struggle to capture the true data-generating process, so a large state space is needed to compensate. The model may also be hard to interpret when using a very large state space and normalising flows. This is in contrast to their traditional use in epidemics/disease progression where specific structures are imposed on transition rate matrix (Pagendam et al., 2020; Ross et al., 2009)). Nonetheless, post analysis on the hidden states still provides insights.

## 6 REPRODUCIBILITY STATEMENT

Datasets used in the paper could be download at the link provided in Appendix A.3 (except LRFF which is a private dataset). We have provided the code we use for data preprocessing and running the experiments in Supplementary Material.

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

# A APPENDIX

## A.1 FAST COMPUTATION WITH KRONECKER PRODUCTS

Algorithm 2 uses the mixed-product property. For two matrices: $\mathbf{A}_i, \mathbf{A}_j \in \mathbb{R}^{2 \times 2}$, we have:

$$\mathbf{A}_i \otimes \mathbf{A}_j = (\mathbf{I}_2 \otimes \mathbf{A}_j)(\mathbf{A}_i \otimes \mathbf{I}_2), \tag{5}$$

where $\mathbf{I}_k$ denotes the $k \times k$ identity matrix. This can be easily generalised to the case with 3 matrices, so that:

$$
\begin{aligned}
&\mathbf{A}_i \otimes \mathbf{A}_j \otimes \mathbf{A}_k \\
=&(\mathbf{I}_4 \otimes \mathbf{A}_k)((\mathbf{A}_i \otimes \mathbf{A}_j) \otimes \mathbf{I}_2) \\
=&(\mathbf{I}_4 \otimes \mathbf{A}_k)(\mathbf{A}_i \otimes (\mathbf{A}_j \otimes \mathbf{I}_2)) \\
=&(\mathbf{I}_4 \otimes \mathbf{A}_k)(\mathbf{I}_2 \otimes \mathbf{A}_j \otimes \mathbf{I}_2)(\mathbf{A}_i \otimes \mathbf{I}_4) \\
=&(\mathbf{I}_4 \otimes \mathbf{A}_k \otimes \mathbf{I}_1)(\mathbf{I}_2 \otimes \mathbf{A}_j \otimes \mathbf{I}_2)(\mathbf{I}_1 \otimes \mathbf{A}_i \otimes \mathbf{I}_4),
\end{aligned}
$$

if we generalise $\mathbf{I}_1$ to represent a scalar 1. The above rules can be applied recursively to obtain:

$$
\begin{aligned}
&\mathbf{A}_1 \otimes \cdots \otimes \mathbf{A}_m \\
=&\prod_{i=1}^{m}(\mathbf{I}_{2^{m-i}} \otimes \mathbf{A}_{m-i+1} \otimes \mathbf{I}_{2^{i-1}}).
\end{aligned}
$$

Then its product with a vector becomes:

$$
\begin{aligned}
&(\mathbf{A}_1 \otimes \cdots \otimes \mathbf{A}_m)\boldsymbol{\alpha}_1 \\
=&\left(\prod_{i=1}^{m}(\mathbf{I}_{2^{m-i}} \otimes \mathbf{A}_{m-i+1} \otimes \mathbf{I}_{2^{i-1}})\right)\boldsymbol{\alpha}_1.
\end{aligned}
$$

We need to calculate $\boldsymbol{\alpha}_{i+1} = (\mathbf{I}_{2^{m-i}} \otimes \mathbf{A}_{m-i+1} \otimes \mathbf{I}_{2^{i-1}})\boldsymbol{\alpha}_i$ until we get $\boldsymbol{\alpha}_{m+1}$. Note that this product can be calculated by properly arranging the elements of $\boldsymbol{\alpha}_i$, leading to the reshape operation in Algorithm 2.

## A.2 DECODING WITH THE VITERBI ALGORITHM

---

**Algorithm 3** Fast Decoding Algorithm for CTHMM.

---

**Input:** Observation $\mathbf{o} = [\mathbf{o}_{t_1}^\top, \ldots, \mathbf{o}_{t_n}^\top]^\top$;
parameters $\boldsymbol{\pi}, \{\mathbf{Q}^{(s)}\}_{s=1}^{m}, \{\boldsymbol{f}_j\}_{j=1}^{\bar{m}}, \{\boldsymbol{\mu}_j, \boldsymbol{\Sigma}_j\}_{j=1}^{2^m}$;
**Output:** Most probable hidden state sequence $\mathbf{h} = [\mathbf{h}_{t_1}^\top, \ldots, \mathbf{h}_{t_n}^\top]^\top$;
probability $p(\mathbf{h}|\mathbf{o})$.
 1: **for** $k = 1$ **to** $n$ **do**
 2:     **for** $j = 1$ **to** $2^m$ **do**
 3:         $\beta_j(\mathbf{o}_{t_k}) = \phi(\boldsymbol{f}_j(\mathbf{o}_{t_k})|\boldsymbol{\mu}_j, \boldsymbol{\Sigma}_j)$;
 4:     **end for**
 5: **end for**
 6: $\boldsymbol{v}(1) = \boldsymbol{\pi} \odot \boldsymbol{\beta}(\mathbf{o}_{t_1})$;
 7: **for** $k = 2$ **to** $n$ **do**
 8:     calculate $\mathbf{P}^{(1)}(\Delta t_k), \ldots, \mathbf{P}^{(m)}(\Delta t_k)$ with (3);
 9:     calculate $\hat{\boldsymbol{v}}(k), \boldsymbol{\omega}(k) = g(\mathbf{P}^{(1)}(\Delta t_k), \ldots, \mathbf{P}^{(m)}(\Delta t_k), \boldsymbol{v}(k-1))$ with Algorithm 4;
10:     $\boldsymbol{v}(k) = \hat{\boldsymbol{v}}(k) \odot \boldsymbol{\beta}(\mathbf{o}_{t_k})$;
11: **end for**
12: $p(\mathbf{h}|\mathbf{o}) = \max_j v_j(n)$.
13: $q_n = \arg\max_j v_j(n), q_{n-1} = \omega_{q_n}(n), \ldots, q_1 = \omega_{q_2}(2)$
14: $\mathbf{h} = [\mathbf{h}_{t_1}^\top, \ldots, \mathbf{h}_{t_n}^\top]^\top$ where $\mathbf{h}_{t_k} = b^{-1}(q_k)$.

---

The decoding of CTHMM produces the most likely hidden state sequence, $\arg\max_{\mathbf{h}_{t_1},\ldots,\mathbf{h}_{t_n}} p(\mathbf{h}_{t_1},\ldots,\mathbf{h}_{t_n}|\mathbf{o})$, given the observation sequence $\mathbf{o}$ and model parameters $\left\{\boldsymbol{\pi},\{\mathbf{Q}^{(s)}\}_{s=1}^m,\{\boldsymbol{f}_j\}_{j=1}^{\bar{m}},\{\boldsymbol{\mu}_j,\boldsymbol{\Sigma}_j\}_{j=1}^{2^m}\right\}$.

The decoding can be achieved through Viterbi algorithm, similar to Algorithm 1 – we use our Scalable CTHMM design to make it faster. Algorithm 3 uses Algorithm 4 in line 9 which is a similar subroutine to Algorithm 2 calculating $\hat{\boldsymbol{v}}(k), \boldsymbol{\omega}(k) = g(\mathbf{P}^{(1)}(\Delta t_k),\ldots,\mathbf{P}^{(m)}(\Delta t_k),\boldsymbol{v}(k-1))$. The operation $\bar{\times}$ in Algorithm 4 resembles vector-matrix multiplication, replacing sum with $\max$, obtaining $\boldsymbol{v}$, and replacing sum with $\arg\max$ obtaining $\mathbf{b}$. It has similar complexity as Algorithm 2 thus being equally efficient.

---

**Algorithm 4** Calculate $\hat{\boldsymbol{v}},\boldsymbol{\omega} = g(\mathbf{P}^{(1)},\ldots,\mathbf{P}^{(m)},\boldsymbol{v})$.

---

**Input:** Matrices $\mathbf{P}^{(1)},\ldots,\mathbf{P}^{(m)} \in \mathbb{R}^{2\times2}$, vector $\boldsymbol{v} \in \mathbb{R}^{2^m}$;
**Output:** $\hat{v},\boldsymbol{\omega}$.

1: $\mathbf{B} = [\,]$
2: **for** $s=1$ **to** $m$ **do**
3: $\quad \boldsymbol{v} = \text{reshape}(\boldsymbol{v},2,2^{m-1})$;
4: $\quad \boldsymbol{v},\mathbf{b} = \boldsymbol{v}^\top \bar{\times} \mathbf{P}^{(s)}$;
5: $\quad \boldsymbol{B} = \text{reshape}(\mathbf{B},s-1,2,2^{m-1})$;
6: $\quad$ concatenate $\mathbf{b}$ to the first dimension of $\mathbf{B}$;
7: **end for**
8: $\boldsymbol{B} = \text{reshape}(\mathbf{B},m,2^m)$;
9: $\hat{\boldsymbol{v}} = \text{flatten}(\boldsymbol{v})$;
10: Transform each row of $\mathbf{B}$ (binary vector) to integer obtaining $\boldsymbol{\omega}$.

---

### A.3 More Details and Complete View of the Datasets

#### A.3.1 Taxi Dataset

The Taxi dataset (Yuan et al., 2010; 2011) contains the GPS trajectories of 10,357 taxis during the period of Feb. 2 to Feb. 8, 2008 within Beijing[1]. The total number of points in this dataset is about 15 million and the total distance of the trajectories reaches to 9 million kilometres. We removed outliers and restricted the GPS trajectories to those with latitude in the interval $[39.50015, 40.29868]$ and longitude in the interval $[116.10023, 116.79973]$. After preprocessing, there were 3,127 taxis and 3,845,410 observed locations.

#### A.3.2 RAATD Dataset

The Retrospective Analysis of Antarctic Tracking Data (RAATD) is a Scientific Committee for Antarctic Research project led jointly by the Expert Groups on Birds and Marine Mammals and Antarctic Biodiversity Informatics (Ropert-Coudert et al., 2020). It includes tracking data from over 70 contributors across 12 national Antarctic programs, and includes data from 17 predator species, 4060 individual animals, and over 2.9 million observed locations. We use the standardised version of the dataset[2] and clean the data to keep 2.5 million observations. The data cover a large area where the Longitude is not continuous from -180 to 180, so we transform all Latitude/Longitude to a 3D-Cartesian coordinates with an approximate altitude of zero and origin at the geographical centre of Earth. In doing so, we model 3D movement data. Observations in this dataset can come with labels associated with time (breeding stage) or other information (taxonomy of the species, sex, age class). This results in four classification tasks that can be used to verify the interpretability of the state space with the Viterbi (decoding) algorithm.

#### A.3.3 LRFF Dataset

The Little red flying foxes (LRFF) contains GPS coordinates of little red flying foxes, *megachiropteran* bats native to northern and eastern Australia. There were 51 individual flying foxes equipped with

---

[1]https://www.kaggle.com/datasets/arashnic/tdriver.
[2]https://data.aad.gov.au/metadata/SCAR_EGBAMM_RAATD_2018_Standardised.

GPS collars which sent noisy signals on their locations intermittently depending on a variety of internal/external conditions. The data was collected over a period of more than 3 years and spread over a wide geographic range of northern and eastern Australia. The study of their movement helps understand their behaviour and avoid eroding their natural habitat.

We present all the observations after a simple data cleaning/preprocessing steps in Fig. 7. The datasets are still noisy and challenging to model.

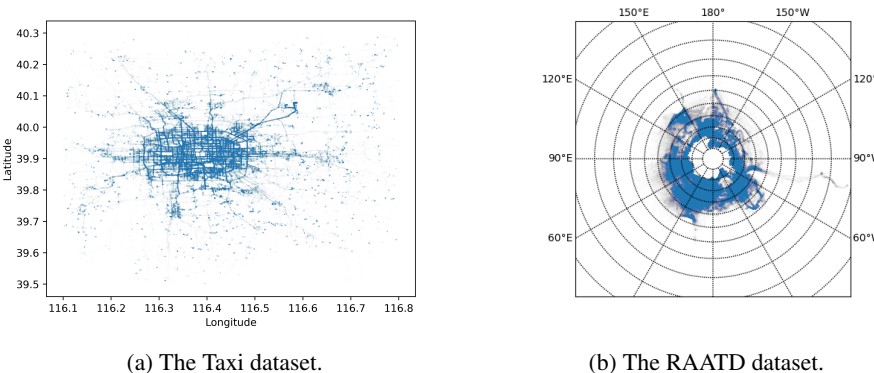

(a) The Taxi dataset.                    (b) The RAATD dataset.

Figure 7: A complete view of the datasets.

## A.4 EVALUATION METRICS

The Negative Log-Likelihood (-LogLik) of $n$ sequences $\mathbf{o}_1, ..., \mathbf{o}_n$ with length $l_1, ..., l_n$ w.r.t. a distribution with probability densities $p(\cdot)$ is defined as $-\frac{\sum_{k=1}^{n} \log p(\mathbf{o}_k)}{\sum_{k=1}^{n} l_k}$.

The CRPS (Continuous Ranked Probability Score) is a proper scoring function used to evaluate probabilistic forecasts by comparing the predicted cumulative distribution function (CDF) against the observations. It is defined as:

$$CRPS(F, x) = \int_{-\infty}^{\infty} \Big( F(y) - \mathbb{1}(y - x) \Big)^2 dy,$$

where $x$ is an observation and $F$ is the CDF associated with the empirical probabilistic forecast. $\mathbb{1}$ is an indicator function which gives 0 with negative inputs or otherwise 1.

In some cases we do not have the closed-from expression of CRPS or $F$, as is in our case. In this case we follow the implementation in pyro (Bingham et al., 2019) to empirically calculate CRPS through sampling following an equivalent formulation:

$$CRPS(F, x) = E[|\hat{X} - x|] - \frac{1}{2} E[|\hat{X} - \hat{X}'|],$$

where $\hat{X}$ and $\hat{X}'$ are independently and identically distributed according to $F$. The sample size was set to 50 per observation. The CRPS we reported are averaged across observations and dimensions.

## A.5 MORE DETAILS ON DOWNSTREAM TASKS

The details of the tasks on the RAATD dataset are:

    i. species classification with 17 different species;

    ii. maturity classification with 4 age classes from different species;

    iii. sex classification with 2 distinct labels;

    iv. breeding stage classification with 17 labels.

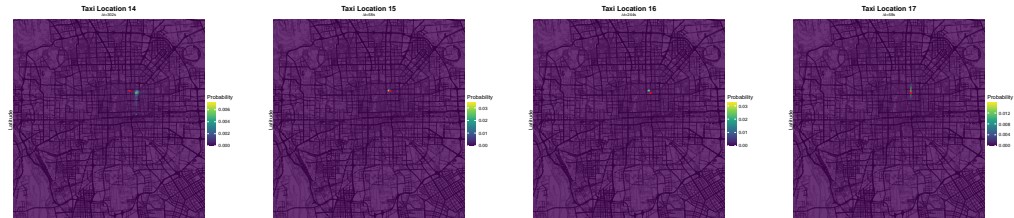

Figure 8: Four consecutively predicted heatmaps by our model on the Taxi dataset (red dots indicate observations).

Note that some of the labels implicitly indicate the taxonomy of species, *e.g.*, some breeding stage labels are only for a specific kind of animals. For the first 3 tasks, the classifier takes as input the 4009 samples generated from averaging the hidden states within each sequence. Task iv includes 380638 labelled observations. For multi-class classification, we adopt "macro" and "ovr" scheme in sklearn (Pedregosa et al., 2011) for the AUC calculation.

### A.5.1 MORE RESULTS

Figure 8 presents a few predicted heatmaps on taxi dataset generated by sampling from our model.

