# OpenReview forum: "Scalable Continuous-Time Hidden Markov Models"
_ICLR.cc/2026/Conference — Submitted to ICLR 2026_

### Official Review · Reviewer_uP49 · 2025-10-31

**Soundness:** 3
**Presentation:** 3
**Contribution:** 2
**Rating:** 4
**Confidence:** 3

**Summary:**

This work considers continuous-time hidden Markov models (CTHMM) whose latent state state at time $t$, $H(t)$, takes values in $\{1, ..., M\} =: [M]$ for some large integer $M$ (this work considers $M \leq 2^{12} = 4096$). Standard recursions for performing inference in such models scale exponentially with $M$ and are thus prohibitively costly.

To circumvent this problem, the authors propose a continuous-time analogue of the factorial HMM. Specifically, assuming that $m := \log_2(M) \in \mathbb{N}$, they specify $m$ independent continuous-time binary-state-space Markov chains  $(H_1(t))$, ..., $(H_m(t))$ and then set
$$
  H(t) := b(((H_1(t), ..., H_m(t)))),
$$
for some suitable bijection $b: \{0,1\}^m \to [M]$. Exploiting the fact that the binary-state-space Markov chains $(H_l(t))$ are thus conditional independent given the observations construction then makes inference in such models possible even if $M$ is large.

To further reduce the computational cost, the authors specify the observation densities/probabilities through $\bar{m}$ normalising flows $f_1, ..., f_{\bar{m}}$, where $\bar{m}$ is much smaller than $M$ (the authors consider $\bar{m} \leq 8$). That is, they specify another mapping $b' : [M] \to [\bar{m}]$ and then assume that the observation density/probability satisfies
$$
  p(o_t|H(t)) = p(o_t|f_{b'(H(t))}, \mu_{H(t)}, \varSigma_{H(t)}).
$$
Here, $\mu_j$ and $\varSigma_j$, for $j \in [M]$, are the parameters of the base distribution for the normalising flow used to parametrise the observation density.

**Strengths:**

This paper is fairly well written and thus easy to read. I particularly appreciate Figures 1 and 2 which make the proposed approach very clear. The idea of extending factorial HMMs to the continuous-time setting seems sensible and is novel to my knowledge. The method outperform state-of-the-art alternatives in the metrics considered and the authors take great care to demonstrate, in the ablation studies, the benefits of different aspects of their methodology in isolation.

**Weaknesses:**

**Main comment:**

Line 357 states that the models were trained on the first 80 % of each observation sequence and then metrics (e.g., the negative log-likelihood) were computed on 100 % of the observation sequence. This leaves a chance that the superior performance of the proposed method stems from overfitting on the first 80 % of the observations.

Thus, I would like the authors to include comparisons of the different methods in terms of out-of-sample predictions for each data set, too. Presumably, prediction is one of the main tasks for employing such methods anyway.



**Minor issues:***

- I would add brackets around the second product and the term immediately following in Equation 1.
- I think it would help the reader to mention much earlier in the paper (e.g. early on in the introduction and maybe even in the abstract) that the proposed method is essentially a continuous-time analogue of the factorial HMM (with a particular type of observation models specified through normalising flows).
- There are some very minor typos (e.g., "compare" in L47, "CTHMM" (i.e., singular), in L44 & L48)
- The text in the panels of Figure 4 is much too small.
- The bibliography has a large number of typos and inconsistencies, e.g., missing capital letters in proper nouns or in journal names, inconsistencies in formatting (abbreviated journal names vs unabbreviated; naming of conferences), some author first names are abbreviated but others aren't, use of "." (or not) in abbreviated author names.

**Questions:**

1. How different are the parameters $\mu_j$ and $\varSigma_j$ for those states that use the same normalising flow? If they are similar, then the proposed model becomes essentially a standard $\bar{m}$-state CTHMM (because only $\bar{m}$ states are then identifiable) with some restrictions placed on the $\bar{m} \times \bar{m}$ transition matrix.

2. Why is does the computational (training) cost of the FaHMM scale so much poorer than the proposed methodology? Don't both approaches exploit the same type of conditional independence structure?

3. What is the observation model used by the standard CTHMM baseline?

---

> ### Author Response · Authors · 2025-11-28
> **Response to main weakness**
>
> Thank you a lot for pointing out the minor issues and we will address them accordingly in our manuscript. Regarding the main weakness and questions, we will answer point by point here.
>
> **Response to the main weakness**
>
> Fitting on all available data can be useful for discovering underlying phenomena/dynamics of the sequences (by examining the transition rate/probability matrix), which is common in biology [1]. Having said that, we report below the -LogLik(CRPS) on the 20\% test data only (conditioned on the observed data) and will put this in appendix of the manuscript. Results show that on the Taxi dataset, increasing hidden states helps model perform better on both train \& test set. While on RAATD and LRFF dataset, over-fitting occurs. Our method may benefit from using separate validation sets in these cases.   Apart from the prediction task,  downstream tasks are also where the value of HMM sits in. The advantage of HMM is mainly on identifying any physical meanings of latent states, for example, to see whether such factorised latent states can be clustered into meaningful groups using all data, and subsequently investigating the transition rate/probability matrix and emission distributions to study animal behaviour.
>
> [1] Krogh, Anders, et al. "Hidden Markov models in computational biology: Applications to protein modeling." Journal of molecular biology 235.5 (1994): 1501-1531.
>
>
> |          | **Taxi**      |               |               |               | **RAATD**     |               |               |               | **LRFF**      |               |               |                |
> | -------: | ------------- | ------------- | ------------- | ------------- | ------------- | ------------- | ------------- | ------------- | ------------- | ------------- | ------------- | -------------- |
> |   **M**  | CTHMM         | HMM           | FaHMM         | Ours          | CTHMM         | HMM           | FaHMM         | Ours          | CTHMM         | HMM           | FaHMM         | Ours           |
> |    **2** | 0.045 (0.014) | 0.046 (0.014) | 0.047 (0.014) | 0.053 (0.014) | 0.060 (0.003) | 0.045 (0.003) | 0.025 (0.001) | 0.028 (0.001) | 0.007 (0.003) | 0.022 (0.003) | 0.025 (0.003) | 0.014 (0.003)  |
> |    **4** | 0.051 (0.012) | 0.059 (0.016) | 0.046 (0.011) | 0.058 (0.013) | 0.054 (0.003) | 0.030 (0.002) | 0.034 (0.001) | 0.040 (0.001) | 0.008 (0.001) | 0.021 (0.001) | 0.071 (0.003)  | -0.001 (3e-5) |
> |    **8** | -             | 0.050 (0.009) | 0.054 (0.009) | 0.053 (0.009) | -             | 0.022 (0.002) | 0.038 (0.002) | 0.087 (0.001) | 0.009 (0.001) | 0.007 (0.002) | 0.001 (0.004) | 0.004 (0.002)  |
> |   **16** | -             | 0.049 (0.007) | 0.049 (0.007) | 0.049 (0.007) | -             | 0.042 (0.003) | 0.044 (0.003) | 0.033 (0.001) | -             | 0.104 (0.003) | 0.120 (0.003) | 0.006 (0.002)  |
> |   **32** | -             | 0.053 (0.006) | 0.049 (0.005) | 0.051 (0.005) | -             | 0.078 (0.003) | 0.046 (0.003) | 0.066 (0.003) | -             | 0.107 (0.003) | 0.127 (0.003) | 0.007 (0.002)  |
> |   **64** | -             | 0.050 (0.004) | 0.044 (0.004) | 0.051 (0.004) | -             | 0.063 (0.004) | 0.059 (0.003) | 0.068 (0.003) | -             | 0.127 (0.003) | 0.091 (0.001) | 0.011 (0.002)  |
> |  **128** | -             | 0.047 (0.003) | 0.042 (0.004) | 0.041 (0.003) | -             | 0.056 (0.002) | 0.082 (0.003) | 0.066 (0.004) | -             | 0.097 (0.004) | 0.107 (0.005) | 0.006 (0.001)  |
> |  **256** | -             | -             | 0.040 (0.004) | 0.038 (0.003) | -             | -             | 0.065 (0.003) | 0.081 (0.004) | -             | 0.123 (0.002) | 0.106 (0.001) | 0.008 (0.002)  |
> |  **512** | -             | -             | 0.041 (0.004) | 0.033 (0.002) | -             | -             | 0.048 (0.003) | 0.072 (0.003) | -             | 0.123 (0.003) | 0.080 (0.004) | 0.010 (0.002)  |
> | **1024** | -             | -             | 0.039 (0.003) | 0.028 (0.002) | -             | -             | 0.055 (0.003) | 0.073 (0.003) | -             | -             | 0.096 (0.003) | 0.016 (0.002)  |
> | **2048** | -             | -             | 0.036 (0.003) | 0.026 (0.002) | -             | -             | 0.072 (0.002) | 0.080 (0.002) | -             | -             | 0.101 (0.006) | 0.005 (0.001)  |
> | **4096** | -             | -             | -             | 0.023 (0.002) | -             | -             | -             | 0.070 (0.009) | -             | -             | 0.077 (0.004) | 0.008 (0.001)  |

---

> ### Author Response · Authors · 2025-11-28
> **Response to questions**
>
> **How different are the parameters and for those states that use the same normalising flow?**
>
> They are learnt by forward algorithm and different enough to be distinguished within the group sharing same normalising flows, or otherwise adding more hidden states will have no effect since we keep using 8 normalising flow models (for \# hidden states >8).
>
>
> **Why is does the computational (training) cost of the FaHMM scale so much poorer than the proposed methodology?**
>
> As a discrete-time model, for predicting multiple time steps ahead (after discretisation of time intervals) FaHMM needs to calculate the transition probability matrices with matrix power and store the intermediate variables for training.  Increasing the size of time steps and number of distinct time interval in a dataset make the model intractable sooner than its continuous variant because calculation of matrix power, which highlights the value of our continuous model which is not affected by the value of time interval (in terms of computation complexity).
>
>
> **What is the observation model used by the standard CTHMM baseline?**
>
> It was multivariate normal distribution (for scalability).

---

### Official Review · Reviewer_ghPA · 2025-11-01

**Soundness:** 2
**Presentation:** 2
**Contribution:** 2
**Rating:** 4
**Confidence:** 3

**Summary:**

This work proposes a strategy to scale Continuous-time Hidden Markov models (CTHMM) using a factorisation of the state space. In particular they model the latent state transition as the evolution of independent Markov processes, with binary state spaces. They then show that the evolution of the collection is follows a transition matrix that can be expressed as the Kronecker product of the transition matrices of the independent processes, which in turn allows fast algorithms through clever use of Kronecker algebra. The loss of expressivity, as a consequence of the independence assumption is counteracted by introducing flexible emission densities through the use of normalising flows. The experimental results some improvement in terms of negative log likelihood metric in comparison to a vanilla HMM, a factorial HMM (both discrete-time HMMs). Furthermore, it was shown that by the virtue of allowing a larger state-space than the competing discrete-time HMMs, the proposed model has better performance in downstream tasks.

**Strengths:**

The major strength of the paper is that it suggests a straightforward approach to have a large state-space within the CTHMM framework. This when combined with a normalising flow (NF) does indeed produce an expressive model. The idea of using parallel independent latent processes coupled with fast Kronecker algebra appears to be novel.

**Weaknesses:**

I believe the major weaknesses stem from the fact that the experiments do not adequately justify the main claim i.e. the usefulness of this model over available alternatives.

Major:
 - As far as I understand, in Table 2 the metrics were obtained on the full sequence (involving the training part as well). A model having a larger state space and a NF can overfit the training data and may come out best in terms of the negative Log Likelihood (NLL) metric. With NLL as a metric (and inclusion of training data) the ablation studies also remain inconclusive, again the improvements can be due to overfitting.
- The downstream task performance perhaps is a bit more clearer indication of the usefulness of having larger state spaces. But here the improvement over a vanilla HMM is not substantial. Especially considering the additional computational cost associated with this method.
- The experiments do not probe the limitations of the independence assumption. How well this model learns the true latent state sequences? It is difficult to scale CTHMM, but the proposed approach of scaling is (in a way) shifting the goal post by introducing a different model. Thus, the authors should have shown how well this "different" model is still able to capture the "same" underlying process.

- The introduction of a NF is not well justified. I mean one can add such a NF easily to a vanilla HMM (and its other derivatives).

Minor:
 - The paper structuring needs improvements. The model should have been specified first and then all ther notations should be introduced in sequence.

- There should be concise description of how the model parameters are learnt, what the forward/Viterbi algorithms are. Without a clear flow from the model to the learning and finally to the inference, the presentation seems haphazard.
- I am not sure whether Figure 2 is adding any value. I had to read the text carefully to understand the NF construction.

**Questions:**

- The real question is whether adding more states (following this approach) provides better performance than other approaches, for standard sequence prediction tasks, without incurring additional computational cost? This aspect is not tested adequately.  It would have been interesting to see a comparison on just unseen trajectory.

- Following Theorem 2.1, one would care to know how well $\hat{\mathbf{Q}}$ approximates $\mathbf{Q}$? The authors need to answer this unambiguously, though either theoretical tools or additional experiments.

- Why was [1] not included in the benchmarking?

- How was $\Delta t_{k}$ chosen?

- How does one come-up with the optimal grouping (of the NF) number?

1. Yu-Ying Liu, Shuang Li, Fuxin Li, Le Song, and James M Rehg. Efficient learning of continuous-time
hidden markov models for disease progression. Advances in neural information processing systems,
28, 2015.

---

> ### Author Response · Authors · 2025-11-28
> **Response to weakness 1 and question 1**
>
> Fitting on all available data can be useful for discovering underlying phenomena/dynamics of the sequences (by examining the transition rate/probability matrix), which is common in biology [1]. Having said that, we report below the -LogLik(CRPS) on the 20\% test data only (conditioned on the observed data) and will put this in appendix of the manuscript. Results show that on the Taxi dataset, increasing hidden states helps model perform better on both train \& test set. While on RAATD and LRFF dataset, over-fitting occurs. Our method may benefit from using separate validation sets in these cases.
>
> [1] Krogh, Anders, et al. "Hidden Markov models in computational biology: Applications to protein modeling." Journal of molecular biology 235.5 (1994): 1501-1531.
>
>
> |          | **Taxi**      |               |               |               | **RAATD**     |               |               |               | **LRFF**      |               |               |                |
> | -------: | ------------- | ------------- | ------------- | ------------- | ------------- | ------------- | ------------- | ------------- | ------------- | ------------- | ------------- | -------------- |
> |   **M**  | CTHMM         | HMM           | FaHMM         | Ours          | CTHMM         | HMM           | FaHMM         | Ours          | CTHMM         | HMM           | FaHMM         | Ours           |
> |    **2** | 0.045 (0.014) | 0.046 (0.014) | 0.047 (0.014) | 0.053 (0.014) | 0.060 (0.003) | 0.045 (0.003) | 0.025 (0.001) | 0.028 (0.001) | 0.007 (0.003) | 0.022 (0.003) | 0.025 (0.003) | 0.014 (0.003)  |
> |    **4** | 0.051 (0.012) | 0.059 (0.016) | 0.046 (0.011) | 0.058 (0.013) | 0.054 (0.003) | 0.030 (0.002) | 0.034 (0.001) | 0.040 (0.001) | 0.008 (0.001) | 0.021 (0.001) | 0.071 (0.003)  | -0.001 (3e-5) |
> |    **8** | -             | 0.050 (0.009) | 0.054 (0.009) | 0.053 (0.009) | -             | 0.022 (0.002) | 0.038 (0.002) | 0.087 (0.001) | 0.009 (0.001) | 0.007 (0.002) | 0.001 (0.004) | 0.004 (0.002)  |
> |   **16** | -             | 0.049 (0.007) | 0.049 (0.007) | 0.049 (0.007) | -             | 0.042 (0.003) | 0.044 (0.003) | 0.033 (0.001) | -             | 0.104 (0.003) | 0.120 (0.003) | 0.006 (0.002)  |
> |   **32** | -             | 0.053 (0.006) | 0.049 (0.005) | 0.051 (0.005) | -             | 0.078 (0.003) | 0.046 (0.003) | 0.066 (0.003) | -             | 0.107 (0.003) | 0.127 (0.003) | 0.007 (0.002)  |
> |   **64** | -             | 0.050 (0.004) | 0.044 (0.004) | 0.051 (0.004) | -             | 0.063 (0.004) | 0.059 (0.003) | 0.068 (0.003) | -             | 0.127 (0.003) | 0.091 (0.001) | 0.011 (0.002)  |
> |  **128** | -             | 0.047 (0.003) | 0.042 (0.004) | 0.041 (0.003) | -             | 0.056 (0.002) | 0.082 (0.003) | 0.066 (0.004) | -             | 0.097 (0.004) | 0.107 (0.005) | 0.006 (0.001)  |
> |  **256** | -             | -             | 0.040 (0.004) | 0.038 (0.003) | -             | -             | 0.065 (0.003) | 0.081 (0.004) | -             | 0.123 (0.002) | 0.106 (0.001) | 0.008 (0.002)  |
> |  **512** | -             | -             | 0.041 (0.004) | 0.033 (0.002) | -             | -             | 0.048 (0.003) | 0.072 (0.003) | -             | 0.123 (0.003) | 0.080 (0.004) | 0.010 (0.002)  |
> | **1024** | -             | -             | 0.039 (0.003) | 0.028 (0.002) | -             | -             | 0.055 (0.003) | 0.073 (0.003) | -             | -             | 0.096 (0.003) | 0.016 (0.002)  |
> | **2048** | -             | -             | 0.036 (0.003) | 0.026 (0.002) | -             | -             | 0.072 (0.002) | 0.080 (0.002) | -             | -             | 0.101 (0.006) | 0.005 (0.001)  |
> | **4096** | -             | -             | -             | 0.023 (0.002) | -             | -             | -             | 0.070 (0.009) | -             | -             | 0.077 (0.004) | 0.008 (0.001)  |

---

> ### Author Response · Authors · 2025-11-28
> **response to weakness 2, weakness 3 and question2, weakness 4, question 3-5**
>
> **weakness2:  The downstream task performance perhaps is a bit more clearer indication of the usefulness of having larger state spaces. But here the improvement over a vanilla HMM is not substantial. Especially considering the additional computational cost associated with this method**
>
> Vanilla HMM is indeed a good method and has been widely use in practice. Its better performance on Task iv (breeding stage classification) can be credited to that change of breeding stage is a rare event. The computation cost of Vanilla HMM is however comparable to our method. It requires calculation of matrix power for applying to irregular data, and is limited by number of distinct time steps and the value of time steps. It cannot handle original time steps and has to resort to discretisation/binning. This also highlights the value of our continuous model which is not affected by the value of time interval (in terms of computation complexity).
>
> **weakness 3 and question 2 on how well approximates**
>
> Unlike rank decomposition of matrix, there is no asymptotic property for Kronecker sum w.r.t to general matrix. Even if we assume the underlying transition rate matrix is known and has such a format, the normalising flows make this learning problem highly unconvex. Therefore, we are not guaranteed to recover a unique transition rate matrix. We can however verify through downstream tasks to see if the hidden states are meaningful (with training and testing data altogether) so that results indicate learning of proper latent states and transition rate matrix, as in Task i-iv.
>
> **weakness 4: The introduction of a NF is not well justified. I mean one can add such a NF easily to a vanilla HMM (and its other derivatives).**
>
> We add experiments w.r.t adding NFs to a vanilla HMM (similar to Lorek et al NeurIPS 2022 [1]), with the adaptation to irregularly observed data. We implement the method up to 32 hidden states ($M=2^5$, for complete training in reasonable time) on taxi dataset for comparison (including evaluation on full sequences and test parts only) and show the results below which will also be included in appendix. It performs slightly better in training which is to be expected given its less restricted structure, but fails to scale up to more than 32 states (also overfits on training set). We add NFs to our method to compensate the restrictive latent space so that we can capture more complicate process. Fig.6 in our paper also shows the advantages of adding NFs by comparing models with/without the NF components.
>
> | **M** | **-LogLik (all data/test only)** | **CRPS (all data/test only) ** |
> | ----: | --------------------- | ------------------ |
> |     2 | 2.020 (0.062)         | 0.486 (0.015)      |
> |     4 | 1.409 (0.060)         | 0.409 (0.015)      |
> |     8 | 0.877 (0.053)         | 0.291 (0.010)      |
> |     16| 0.366 (0.050)         | 0.215 (0.008)      |
> |     32| -0.248 (0.055)        | 0.154 (0.005)      |
>
> [1] Lorek, Pawel, et al. "FlowHMM: Flow-based continuous hidden Markov models." Advances in Neural Information Processing Systems 35 (2022): 8773-8784.
>
> **question3: Why was [1] not included in the benchmarking?**
>
> [1] requires a predefined/restricted structure of transition rate matrix (some prior knowledge according to specific application), and/or a data emission model. It needs a dataset with few distinct time intervals (e.g. 3 and 63 in the paper). It is therefore not feasible for general datasets as in our paper (and their code is not available).
>
> [1] Yu-Ying Liu, Shuang Li, Fuxin Li, Le Song, and James M Rehg. Efficient learning of continuous-time hidden markov models for disease progression. Advances in neural information processing systems, 28, 2015.
>
> **question4 - How was $\Delta t_k$ chosen? and question 5 - How does one come-up with the optimal grouping (of the NF) number?**
>
>
> $\Delta t_k$ is a variable calculated from neighbouring timestamps (e.g. $t_{i+1}-t_i$). The optimal grouping number was chosen so that it has enough power to fit data while still scales up to largest dataset (taxi) in our experiments.  We chose a reasonable large (8) number of the normalising flow models for speed and expressiveness trade-off. The next option will be 16 because the number of hidden states needs to be divisible by the number of the normalising flow models (for efficient computation). We present the results with 0,2,4,8 flows on taxi dataset below and will add this to appendix. Generally speaking, more flows will make training and convergence harder but gives better results.
>
> |        \#flows    | **0**          | **2**          | **4**          | **8**          |
> | ----------------- | -------------- | -------------- | -------------- | -------------- |
> | **-LogLik(CRPS)** | -1.163 (0.098) | -1.231 (0.104) | -1.224 (0.102) | -1.315 (0.097) |

---

### Official Review · Reviewer_5jc1 · 2025-11-02

**Soundness:** 2
**Presentation:** 4
**Contribution:** 3
**Rating:** 4
**Confidence:** 3

**Summary:**

This paper improves Continuous Time Hidden Markov Models (CTHMM) by addressing their computational and modeling limitations. First, the authors decompose the hidden state space with M values into $\log_2(M)$ independent binary Markov processes. This structure allows transition probabilities to be computed in closed form and evaluated in parallel, avoiding the high cost and instability of large matrix exponentiations. The independence of each chain limits the expressiveness of the model. To counter this, authors replace simple emission models such as Gaussians or Mixture of Gaussians with conditional normalizing flows, which can represent more complex observation distributions. These changes make CTHMMs more practical for large datasets with irregular time sampling, enabling training in reasonable time. The approach achieves higher likelihoods and better downstream performance than vanilla CTHMM and discrete-time HMM models.

**Strengths:**

- The authors propose improvements that significantly enhance the scalability of CTHMMs.
- The method achieves superior results compared to standard CTHMMs and discrete-time HMMs reported in the literature.
- The manuscript is clearly written and easy to follow.

**Weaknesses:**

- Lines 356 and 454: "Models are trained on the first 80% of each sequence (…) and evaluated on each complete sequence." This train–test split, used across all experiments, does not provide a reliable measure of generalization because 80% of the test data overlaps with the training sequences. This issue is particularly critical for downstream tasks, where the classifier may simply memorize hidden-state patterns from the first 80% of each sequence rather than learning class-level features.
- Line 447: "For each method, we use the trained model with the largest number of hidden states." While this allows comparing models at their limits, it makes it unclear what drives the performance differences — the use of normalizing flows or the larger latent space dimensionality.
- The authors introduce up to eight independent normalizing flow models for grouped latent states, but no ablation studies are provided to justify this design choice.
- Similarly, different Gaussian parameters are used for each latent state as the prior distribution for the normalizing flow, yet there is no evidence showing that this design improves performance. Since the flow is already conditioned on the hidden state, using separate priors may not be necessary.

**Questions:**

- (Ad W1) Would it be possible to validate the models only on the remaining 20% of each sequence (unseen during training)? Alternatively, the authors could train on K% of complete sequences and evaluate on the remaining (100–K)% of entirely unseen sequences to better assess generalization.
- (Ad W2) Table 2 perfectly shows the effect of varying latent dimensionality for each model. Could a similar analysis be performed for the downstream tasks? For example, Figure 5 could be split into four subfigures (one per task) to compare methods directly (rather than one method for each of tasks).
- Minor: In Equation (1), the variable "n" is not defined explicitly and it has to be deduced from the context. Clarifying it in the text would improve readability.

---

> ### Author Response · Authors · 2025-11-28
> **Response to Weakness 1 and (Ad W1)**
>
> Thanks for comments. We will address them point by point.
>
> **Weakness 1 and (Ad W1)**
>
> Fitting on all available data can be useful for discovering underlying phenomena/dynamics of the sequences (by examining the transition rate/probability matrix), which is common in biology [1]. Having said that, we report below the -LogLik(CRPS) on the 20\% test data only (conditioned on the observed data) and will put this in appendix of the manuscript. Results show that on the Taxi dataset, increasing hidden states helps model perform better on both train \& test set. While on RAATD and LRFF dataset, over-fitting occurs. Our method may benefit from using separate validation sets in these cases.  For experiments on downstream tasks, the purpose is mainly on identifying any physical meanings of latent states, to see whether such factorised latent states can be clustered into meaningful groups using all data, and subsequently investigating the transition rate/probability matrix and emission distributions to study animal behaviour.
>
> [1] Krogh, Anders, et al. "Hidden Markov models in computational biology: Applications to protein modeling." Journal of molecular biology 235.5 (1994): 1501-1531.
>
> |          | **Taxi**      |               |               |               | **RAATD**     |               |               |               | **LRFF**      |               |               |                |
> | -------: | ------------- | ------------- | ------------- | ------------- | ------------- | ------------- | ------------- | ------------- | ------------- | ------------- | ------------- | -------------- |
> |   **M**  | CTHMM         | HMM           | FaHMM         | Ours          | CTHMM         | HMM           | FaHMM         | Ours          | CTHMM         | HMM           | FaHMM         | Ours           |
> |    **2** | 0.045 (0.014) | 0.046 (0.014) | 0.047 (0.014) | 0.053 (0.014) | 0.060 (0.003) | 0.045 (0.003) | 0.025 (0.001) | 0.028 (0.001) | 0.007 (0.003) | 0.022 (0.003) | 0.025 (0.003) | 0.014 (0.003)  |
> |    **4** | 0.051 (0.012) | 0.059 (0.016) | 0.046 (0.011) | 0.058 (0.013) | 0.054 (0.003) | 0.030 (0.002) | 0.034 (0.001) | 0.040 (0.001) | 0.008 (0.001) | 0.021 (0.001) | 0.071 (0.003)  | -0.001 (3e-5) |
> |    **8** | -             | 0.050 (0.009) | 0.054 (0.009) | 0.053 (0.009) | -             | 0.022 (0.002) | 0.038 (0.002) | 0.087 (0.001) | 0.009 (0.001) | 0.007 (0.002) | 0.001 (0.004) | 0.004 (0.002)  |
> |   **16** | -             | 0.049 (0.007) | 0.049 (0.007) | 0.049 (0.007) | -             | 0.042 (0.003) | 0.044 (0.003) | 0.033 (0.001) | -             | 0.104 (0.003) | 0.120 (0.003) | 0.006 (0.002)  |
> |   **32** | -             | 0.053 (0.006) | 0.049 (0.005) | 0.051 (0.005) | -             | 0.078 (0.003) | 0.046 (0.003) | 0.066 (0.003) | -             | 0.107 (0.003) | 0.127 (0.003) | 0.007 (0.002)  |
> |   **64** | -             | 0.050 (0.004) | 0.044 (0.004) | 0.051 (0.004) | -             | 0.063 (0.004) | 0.059 (0.003) | 0.068 (0.003) | -             | 0.127 (0.003) | 0.091 (0.001) | 0.011 (0.002)  |
> |  **128** | -             | 0.047 (0.003) | 0.042 (0.004) | 0.041 (0.003) | -             | 0.056 (0.002) | 0.082 (0.003) | 0.066 (0.004) | -             | 0.097 (0.004) | 0.107 (0.005) | 0.006 (0.001)  |
> |  **256** | -             | -             | 0.040 (0.004) | 0.038 (0.003) | -             | -             | 0.065 (0.003) | 0.081 (0.004) | -             | 0.123 (0.002) | 0.106 (0.001) | 0.008 (0.002)  |
> |  **512** | -             | -             | 0.041 (0.004) | 0.033 (0.002) | -             | -             | 0.048 (0.003) | 0.072 (0.003) | -             | 0.123 (0.003) | 0.080 (0.004) | 0.010 (0.002)  |
> | **1024** | -             | -             | 0.039 (0.003) | 0.028 (0.002) | -             | -             | 0.055 (0.003) | 0.073 (0.003) | -             | -             | 0.096 (0.003) | 0.016 (0.002)  |
> | **2048** | -             | -             | 0.036 (0.003) | 0.026 (0.002) | -             | -             | 0.072 (0.002) | 0.080 (0.002) | -             | -             | 0.101 (0.006) | 0.005 (0.001)  |
> | **4096** | -             | -             | -             | 0.023 (0.002) | -             | -             | -             | 0.070 (0.009) | -             | -             | 0.077 (0.004) | 0.008 (0.001)  |

---

> ### Author Response · Authors · 2025-11-28
> **Response to Weakness 2 and (Ad W2)**
>
> Thanks for the advice. We have included more results (AUC(ACC)) for other models per task w.r.t number of latent states below and will organise them into figures in appendix. The results show similar trends (larger latent states gives better performance, and our model are better in 3 out of 4 tasks).
>
> Task i
> | M     | CTHMM         | HMM           | FaHMM         | Ours          |
> | ----: | ------------- | ------------- | ------------- | ------------- |
> |     2 | 0.532 (0.198) | 0.631 (0.254) | 0.500 (0.117) | 0.563 (0.117) |
> |     4 | 0.578 (0.232) | 0.761 (0.376) | 0.668 (0.308) | 0.717 (0.340) |
> |     8 | —             | 0.772 (0.410) | 0.780 (0.287) | 0.792 (0.297) |
> |     16| —             | 0.821 (0.435) | 0.804 (0.348) | 0.826 (0.361) |
> |     32| —             | 0.910 (0.547) | 0.848 (0.413) | 0.867 (0.464) |
> |     64| —             | 0.939 (0.604) | 0.850 (0.400) | 0.873 (0.421) |
> |    128| —             | 0.946 (0.638) | 0.879 (0.434) | 0.905 (0.488) |
> |   256 | —             | —             | 0.872 (0.424) | 0.886 (0.479) |
> |   512 | —             | —             | 0.876 (0.406) | 0.891 (0.394) |
> |  1024 | —             | —             | 0.856 (0.462) | 0.856 (0.448) |
> |  2048 | —             | —             | 0.963 (0.759) | 0.964 (0.674) |
> |  4096 | —             | —             | —             | 0.982 (0.822) |
>
> Task ii
> | M     | CTHMM         | HMM           | FaHMM         | Ours          |
> | ----: | ------------- | ------------- | ------------- | ------------- |
> |     2 | 0.503 (0.833) | 0.595 (0.833) | 0.500 (0.833) | 0.530 (0.833) |
> |     4 | 0.623 (0.833) | 0.763 (0.833) | 0.794 (0.829) | 0.816 (0.833) |
> |     8 | —             | 0.741 (0.861) | 0.802 (0.834) | 0.814 (0.832) |
> |     16 | —             | 0.820 (0.845) | 0.829 (0.856) | 0.834 (0.862) |
> |     32 | —             | 0.889 (0.856) | 0.862 (0.806) | 0.874 (0.839) |
> |     64 | —             | 0.952 (0.875) | 0.853 (0.837) | 0.853 (0.837) |
> |     128 | —             | 0.785 (0.832) | 0.882 (0.860) | 0.886 (0.859) |
> |     256 | —             | —             | 0.810 (0.837) | 0.810 (0.837) |
> |     512 | —             | —             | 0.833 (0.831) | 0.861 (0.848) |
> |    1024 | —             | —             | 0.836 (0.831) | 0.907 (0.850) |
> |    2048 | —             | —             | 0.858 (0.843) | 0.925 (0.889) |
> |    4096 | —             | —             | —             | 0.941 (0.875) |
>
>
> Task iii
> | M     | CTHMM         | HMM           | FaHMM         | Ours          |
> | ----: | ------------- | ------------- | ------------- | ------------- |
> |     2 | 0.516 (0.665) | 0.505 (0.665) | 0.500 (0.665) | 0.572 (0.665) |
> |     4 | 0.530 (0.665) | 0.634 (0.683) | 0.553 (0.672) | 0.657 (0.678) |
> |     8 | —             | 0.610 (0.685) | 0.529 (0.667) | 0.534 (0.669) |
> |     16 | —             | 0.603 (0.680) | 0.594 (0.685) | 0.667 (0.685) |
> |     32 | —             | 0.726 (0.695) | 0.694 (0.690) | 0.725 (0.712) |
> |     64 | —             | 0.763 (0.724) | 0.697 (0.685) | 0.699 (0.695) |
> |     128 | —             | 0.762 (0.736) | 0.632 (0.690) | 0.686 (0.684) |
> |     256 | —             | —             | 0.694 (0.694) | 0.702 (0.691) |
> |     512 | —             | —             | 0.696 (0.692) | 0.704 (0.673) |
> |    1024 | —             | —             | 0.637 (0.696) | 0.689 (0.690) |
> |    2048 | —             | —             | 0.707 (0.692) | 0.765 (0.740) |
> |    4096 | —             | —             | —             | 0.793 (0.750) |
>
>
> Task iv
> | M     | CTHMM         | HMM           | FaHMM         | Ours          |
> | ----: | ------------- | ------------- | ------------- | ------------- |
> |     2 | 0.558 (0.388) | 0.707 (0.659) | 0.669 (0.460) | 0.694 (0.460) |
> |     4 | 0.641 (0.382) | 0.814 (0.731) | 0.809 (0.668) | 0.810 (0.674) |
> |     8 | —             | 0.783 (0.723) | 0.879 (0.743) | 0.883 (0.743) |
> |     16 | —             | 0.905 (0.733) | 0.837 (0.683) | 0.838 (0.684) |
> |     32 | —             | 0.915 (0.764) | 0.924 (0.777) | 0.925 (0.777) |
> |     64 | —             | 0.956 (0.802) | 0.920 (0.775) | 0.928 (0.774) |
> |     128 | —             | 0.966 (0.810) | 0.924 (0.768) | 0.926 (0.768) |
> |     256 | —             | —             | 0.917 (0.753) | 0.919 (0.755) |
> |     512 | —             | —             | 0.870 (0.754) | 0.873 (0.754) |
> |    1024 | —             | —             | 0.851 (0.664) | 0.855 (0.666) |
> |    2048 | —             | —             | 0.877 (0.691) | 0.923 (0.759) |
> |    4096 | —             | —             | —             | 0.941 (0.768) |

---

> ### Author Response · Authors · 2025-11-28
> **Response to weakness 3 and 4**
>
> **weakness3: The authors introduce up to eight independent normalizing flow models for grouped latent states, but no ablation studies are provided to justify this design choice.**
>
> We chose a reasonable large (8 or \# hidden states whichever smaller) number of the normalising flow models for speed and expressiveness trade-off. The next option will be 16 because the number of hidden states needs to be divisible by the number of the normalising flow models. We present the results here with 0,2,4,8 flows (on 256 hidden states for completing the run within reasonable time) on taxi dataset below and will add this to appendix. Generally speaking, more flows will make training and convergence harder but gives better results.
>
>
> |        \#flows    | **0**          | **2**          | **4**          | **8**          |
> | ----------------- | -------------- | -------------- | -------------- | -------------- |
> | **-LogLik(CRPS)** | -1.163 (0.098) | -1.231 (0.104) | -1.224 (0.102) | -1.315 (0.097) |
>
>
> **weakness4:
> Similarly, different Gaussian parameters are used for each latent state as the prior distribution for the normalizing flow, yet there is no evidence showing that this design improves performance. Since the flow is already conditioned on the hidden state, using separate priors may not be necessary.**
>
> Because we are sharing flows among groups of hidden states, different Gaussian parameters are required to distinguish between hidden states, or we end up with same emission distributions within each group, making hidden states therein redundant.

---

### Official Review · Reviewer_hUJA · 2025-11-04

**Soundness:** 2
**Presentation:** 4
**Contribution:** 2
**Rating:** 4
**Confidence:** 4

**Summary:**

This paper develops methods to make continuous time HMMs (CT-HMMs) scale to large state spaces. The proposed approach has two key steps, diagrammed in Fig 1. First, the usual 2^m x 2^m transition rate matrix is factorized into many 2x2 matrices, that each govern one of m separate hidden binary variables with factorization similar to a Factorial-HMM (Ghahramani & Jordan '97). Then, the emission model uses a conditional normalizing flow (Dinh et al. '17) to produce a density over observed vectors at each time stamp, which depends on the current state of the m binary variables.

The first assumption, which assumes m binary variables make up the state at each time, is a simplification of the usual CT-HMM which allows more general Markov distributions over the 2^m possible states. However, the computational advantages of this simplification are argued to be worthwhile, especially since the flexible normalizing flow emission model can hopefully make up for any loss in expressivity of the latent state model.

For each of the 2^m hidden state configurations, indexed by j, there is learned a separate mean vector $\mu_j$ and covariance $\Sigma_j$. However, a state-specific normalizing flow might be too expensive, so instead only $\tilde{m} < 2^m$ distinct normalizing flows are learned, and shared to all $2^m$ states via modular arithmetic indexing (the set of every $\tilde{m}$-th states in index order share a normalizing flow model, but have distinct emission models).

Experiments evaluate this CT-HMM approach against classic CT-HMM methods as well as discrete time HMMs with Gaussian likelihoods and factorized versions of those.

**Strengths:**

I think there's plenty to like about this paper.

* Focus on irregular time series modeling, a problem with lots of practical applications but also elegant theory/methods

* The presented method's runtime speedup is a neat and clever achievement. Avoids standard runtime cost of evaluating CT-HMM likelihood that is cubic O(M^3) in number of states M. Instead, proposed factorization achieves O(M log M)

* Use of normalizing flows is welcome as an engineering win to get flexible emissions

* Experiments demonstrating that they can fit models with 2^12 = 4096 states on real datasets


### Notes on Novelty

For me the key innovation here is the clever use of binary factorization with the efficient evaluation in Algorithm 2 to get runtime speed-up from cubic to "linearithmic".

The use of conditional normalizing flows as an emission model is interesting, but the usage here is a bit "off the shelf" and I don't see new methods or insights about CNFs that were not already widely known (e.g. see how another paper by Lorek et al. 2022 uses CNFs as an emission density for discrete time HMMs).

**Weaknesses:**

# Weaknesses

Here's a brief summary of the issues I see with current manuscript, that prevent me from giving a higher rating at present:

* C1: Need better clarity about likelihood of o_t given hidden state
* E1: Assessment seems to be mostly on training data, not proper generalization
* E2: Is it fair to compare continuous and discrete time likelihoods?
* E3: Missing comparisons to important alternative flexible probabilistic models of irregular time
* E4: Convergence of training needs to be verified
* E5: Focus only on 2-dimensional datasets is a limitation


## Clarity issues

### C1: Clarity about likelihood of o_t given hidden state

There seems to be to conflicting formulas:

* in line 164, the log likelihood $\log p(o_t | b(H(t)) = j)$ is given as a log Normal PDF plus a sum of log determinants
* in line 3 of Alg 1, this same likelihood is given using only the log Normal PDF evaluation, without any determinants

I'm pretty sure the determinants are needed to do the change of variables correctly... so is there something missing in Alg 1?

## Experimental design issues/questions

### E1: Assessment seems to be mostly on training data, not proper generalization

In the experiment descriptions, I see the text

> Models are trained on the first 80% of each sequence to minimise average negative log-likelihood (per observation) and evaluated on each complete sequences

So are reported likelihoods on the full 100% of each sequence? If that's true, this seems like you are mostly assessing *training* quality, not generalization quality.

Typically, models are empirically measured to assess generalization (to new sequences, or to parts of sequences unseen in training). I don't know if the ranking of methods based on this "complete sequence" likelihood is measuring the right thing.

Is the CRPS measure also assessed on "complete sequences"? if so, the same concern applies.

### E2: Is it fair to compare continuous and discrete time likelihoods?

I don't think the main paper provides enough reproducible details about how the provided data with irregular timings was discretized. The paper says:

> discretise irregular intervals by computing mean time differences across the data, normalising, and rounding to integers

So, I first compute the mean $\delta t$ across the dataset, but then how do I normalize? Divide by the empirical standard deviation? Something else? Then I take the transformed time steps and round to the nearest integer? I think an example (worked out in the supplement) would help here.

Imagine a case where some adjacent measurements are quite close to one another, but the normalized mean is much larger. Would these ever get grouped into the same timestep? I can imagine this is possible if both measurements round to the same integer.

I worry if the latter occurs (one discrete time interval holds multiple measurements), that the quantities being evaluated in CT vs discrete time (DT) models are no longer apples-to-apples. We need more certainty that the evaluations in Table 1 are "apples-to-apples" fair.

**Related issue:** the compared DT-HMMs in experiments use only Gaussian likelihoods. I'd expect a comparison to the CNF-based likelihoods for DT-HMMs, using e.g. methods from Lorek et al NeurIPS 2022.

### E3: Missing comparisons to important alternative flexible probabilistic models of irregular time

I appreciate the focus on CT-HMMs and close relatives, but I think for a venue like ICLR most folks in the audience will wonder "how does this compare to other probabilistic models of irregular time series"?

For example, you could consider:
* Schirmer et al's continuous recurrent units: https://proceedings.mlr.press/v162/schirmer22a.html
* work on CNFs for irregular time series by Yalavarthi et al. at AAAI 2025: https://ojs.aaai.org/index.php/AAAI/article/view/35494
* GRU-ODE-Bayes from NeurIPS 2019: https://proceedings.neurips.cc/paper/2019/hash/455cb2657aaa59e32fad80cb0b65b9dc-Abstract.html
* work on stochastic ODEs
* work on scalable Gaussian processes (GPs)

I'd expect to see at least 1 or 2 models not in the CT-HMM or DT-HMM family for comparison on a dataset or two. To be clear, I don't necessarily need the present paper's method to beat this alternative, just for a fair evaluation so the audience can understand the strengths/disadvantages of the present approach.

Given the complex landscape of irregular time models available, now, this experiment is important for establishing the significance of this present paper.


### E4: Did runs converge? Need to verify

I see that all methods were run for 100 epochs, or a 160 hour limit. I respect the time limit for its practicality. But for runs that terminated after 100 epochs, were there sanity checks that convergence occurred?

### E5: Focus on two-dimensional datasets is a limitation

All 3 tested datasets focus on time series that describe

> two-dimensional geographic locations data over extensive spatial areas

This focus on 2D data is a limitation. Many time series of interest to CT-HMM users, such as patient vital signs over time, have much greater dimensionality than 2D.

Is there a reason to think the presented approach wouldn't scale well to higher dimensions? Perhaps estimating the state-specific covariance $\Sigma_j$ becomes difficult when $D$ gets large and there are many states?

**Questions:**

Answering the questions raised above, especially those in E1-E5, are probably most important for me to consider raising my score.

I also have this general question: Is the speedup obtained via the efficient factorization into binary states plus careful evaluation of Kronecker products in Alg 2 here exclusive to the *continuous-time* HMM setting? Would any part of this possibly accelerate discrete time HMMs with factorized binary states?

**Details Of Ethics Concerns:**

No concerns here.

---

> ### Author Response · Authors · 2025-11-27
> **Response to E1**
>
> We thank the reviewer for the valuable comments. We are glad that the reviewer think ``there's plenty to like about this paper''. To the issues (E1-E5) that raised in the review, we will address point by point here (thanks for pointing out C1 and we will fix it accordingly):
>
> E1: Assessment seems to be mostly on training data, not proper generalization:
>
> Fitting on all available data can be useful for discovering underlying phenomena/dynamics of the sequences (by examining the transition rate/probability matrix), which is common in biology [1]. Having said that, we report below the -LogLik(CRPS) on the 20\% test data only (conditioned on the observed data) and will put this in appendix of the manuscript. Results show that on the Taxi dataset, increasing hidden states helps model perform better on both train \& test set. While on RAATD and LRFF dataset, over-fitting occurs. Our method may benefit from using separate validation sets in these cases.
>
> [1] Krogh, Anders, et al. "Hidden Markov models in computational biology: Applications to protein modeling." Journal of molecular biology 235.5 (1994): 1501-1531.
>
>
> |          | **Taxi**      |               |               |               | **RAATD**     |               |               |               | **LRFF**      |               |               |                |
> | -------: | ------------- | ------------- | ------------- | ------------- | ------------- | ------------- | ------------- | ------------- | ------------- | ------------- | ------------- | -------------- |
> |   **M**  | CTHMM         | HMM           | FaHMM         | Ours          | CTHMM         | HMM           | FaHMM         | Ours          | CTHMM         | HMM           | FaHMM         | Ours           |
> |    **2** | 0.045 (0.014) | 0.046 (0.014) | 0.047 (0.014) | 0.053 (0.014) | 0.060 (0.003) | 0.045 (0.003) | 0.025 (0.001) | 0.028 (0.001) | 0.007 (0.003) | 0.022 (0.003) | 0.025 (0.003) | 0.014 (0.003)  |
> |    **4** | 0.051 (0.012) | 0.059 (0.016) | 0.046 (0.011) | 0.058 (0.013) | 0.054 (0.003) | 0.030 (0.002) | 0.034 (0.001) | 0.040 (0.001) | 0.008 (0.001) | 0.021 (0.001) | 0.071 (0.003)  | -0.001 (3e-5) |
> |    **8** | -             | 0.050 (0.009) | 0.054 (0.009) | 0.053 (0.009) | -             | 0.022 (0.002) | 0.038 (0.002) | 0.087 (0.001) | 0.009 (0.001) | 0.007 (0.002) | 0.001 (0.004) | 0.004 (0.002)  |
> |   **16** | -             | 0.049 (0.007) | 0.049 (0.007) | 0.049 (0.007) | -             | 0.042 (0.003) | 0.044 (0.003) | 0.033 (0.001) | -             | 0.104 (0.003) | 0.120 (0.003) | 0.006 (0.002)  |
> |   **32** | -             | 0.053 (0.006) | 0.049 (0.005) | 0.051 (0.005) | -             | 0.078 (0.003) | 0.046 (0.003) | 0.066 (0.003) | -             | 0.107 (0.003) | 0.127 (0.003) | 0.007 (0.002)  |
> |   **64** | -             | 0.050 (0.004) | 0.044 (0.004) | 0.051 (0.004) | -             | 0.063 (0.004) | 0.059 (0.003) | 0.068 (0.003) | -             | 0.127 (0.003) | 0.091 (0.001) | 0.011 (0.002)  |
> |  **128** | -             | 0.047 (0.003) | 0.042 (0.004) | 0.041 (0.003) | -             | 0.056 (0.002) | 0.082 (0.003) | 0.066 (0.004) | -             | 0.097 (0.004) | 0.107 (0.005) | 0.006 (0.001)  |
> |  **256** | -             | -             | 0.040 (0.004) | 0.038 (0.003) | -             | -             | 0.065 (0.003) | 0.081 (0.004) | -             | 0.123 (0.002) | 0.106 (0.001) | 0.008 (0.002)  |
> |  **512** | -             | -             | 0.041 (0.004) | 0.033 (0.002) | -             | -             | 0.048 (0.003) | 0.072 (0.003) | -             | 0.123 (0.003) | 0.080 (0.004) | 0.010 (0.002)  |
> | **1024** | -             | -             | 0.039 (0.003) | 0.028 (0.002) | -             | -             | 0.055 (0.003) | 0.073 (0.003) | -             | -             | 0.096 (0.003) | 0.016 (0.002)  |
> | **2048** | -             | -             | 0.036 (0.003) | 0.026 (0.002) | -             | -             | 0.072 (0.002) | 0.080 (0.002) | -             | -             | 0.101 (0.006) | 0.005 (0.001)  |
> | **4096** | -             | -             | -             | 0.023 (0.002) | -             | -             | -             | 0.070 (0.009) | -             | -             | 0.077 (0.004) | 0.008 (0.001)  |

---

> ### Author Response · Authors · 2025-11-28
> **Response to E2 - E5**
>
> **E2: Is it fair to compare continuous and discrete time likelihoods?**
>
> By "normalise" we mean to divide the time intervals by the mean $\delta t$. Then the time intervals are rounded to the nearest integers. This step will combine some adjacent timestamps to reduce the number of distinct time intervals (say $n$). This is important because we need to calculate and store $n$ distinct transition probability matrix calculated by matrix powers, which is prohibitive if we made $n$ too large. Note that the observations will not be grouped and transition probability matrix for overlapped timestamps will be an identity matrix. For the size of datasets studied in this paper, the discretisation step has been chosen carefully to make fair comparison with continuous models, by reducing such overlaps while keeping a manageable computation complexity.
>
> For CNF-base DT-HMMs e.g. Lorek et al NeurIPS 2022 [1], the method and code provided there only supports single sequence, and up to 4 latent states. We re-implement the method up to 32 hidden states ($M=2^5$) on taxi dataset for comparison (including evaluation on full sequences and test parts only) and show the results below which will also be included in appendix. It performs slightly better in training which is to be expected given its less restricted structure, but fails to scale up to more than 32 states (also overfits on training set).
>
> | **M** | **-LogLik(full/test)** | **CRPS(full/test)** |
> | ----: | --------------------- | ------------------ |
> |     2 | 2.020 (0.062)         | 0.486 (0.015)      |
> |     4 | 1.409 (0.060)         | 0.409 (0.015)      |
> |     8 | 0.877 (0.053)         | 0.291 (0.010)      |
> |     16| 0.366 (0.050)         | 0.215 (0.008)      |
> |     32| -0.248 (0.055)        | 0.154 (0.005)      |
>
> [1] Lorek, Pawel, et al. "FlowHMM: Flow-based continuous hidden Markov models." Advances in Neural Information Processing Systems 35 (2022): 8773-8784.
>
>
> **E3: Missing comparisons to important alternative flexible probabilistic models of irregular time**
>
> Thanks for the heads up on the related work. Following Yalavarthi et al. at AAAI 2025 [2], we add comparison to ProFITi [2], continuous recurrent units (CRU), GRU-ODE, HETVAE, Gaussian Process Regression (GPR) and other related methods compared in [2]. For fair of comparison, we refer to the code provided in [2] and use same dataset partitions/preprocessing for the physionet2012 dataset, and same calculation of CRPS therein. Because our model cannot naively handle asynchronously sampled data (we can, but that will require some additional sampling process), we fill the missing values with a indicator value (zero), and compare with baselines on CRPS on visible values only.  We will add following table and references in the paper. Our method performs significantly better in CRPS thanks to a more consistent modelling process across the dataset and more efficient learning algorithm than baselines , but fall short in handling missing data efficiently.
>
> Physionet’12
> |          | **HETVAE**  | **GRU-ODE** | **Neural-flows** | **CRU**     | **ProFITi** | **Ours**   |
> | -------- | ----------- | ----------- | ---------------- | ----------- | ----------- | ---------- |
> | **CRPS** | 0.278±0.001 | 0.278±0.001 | 0.277±0.003      | 0.363±0.002 | 0.253±0.001 | 0.093±0.002 |
>
> [2] Yalavarthi, Vijaya Krishna, et al. "Probabilistic Forecasting of Irregularly Sampled Time Series with Missing Values via Conditional Normalizing Flows." Proceedings of the AAAI Conference on Artificial Intelligence. Vol. 39. No. 20. 2025.
>
>
> **E4: Did runs converge? Need to verify**
>
> For all methods we adjust the learning rate to make sure steadily decrease of the training loss, and all models converge within 100 epochs (except CTHMM with stop criterion given by the software package). We will explain in details in appendix.
>
>
> **E5: Focus on two-dimensional datasets is a limitation**
>
> You are right about the bottle neck being state-specific covariance/mean and the limit scalability of Normalising Flows to higher dimensionality. We can still handle dozens of features easily (as for physionet2012, 37 features in total). We did include a 3D dataset (RAATD), although it was originally 2D in latitude and longitude. We include the details in appendix: ``The data cover a large area where the Longitude is not continuous from -180 to 180, so we transform all Latitude/Longitude to a 3D-Cartesian coordinates with an approximate altitude of zero and origin at the geographical centre of Earth. In doing so, we model 3D movement data.''
>
> **The general question**
>
> Yes, they can also be applied to accelerate discrete time HMMs.

---

### Author Response · Authors · 2025-12-04

Dear AC,

We have spend a considerable amount of time addressing every concern raised by the reviewers. Given the recent OpenReview API Security Incident, there is no chance for the reviewers to change their scores since our rebuttal was submitted close to time of the incident, despite some positive comments. We hope the positive comments and our detailed response could be taken into consideration.

Specifically, Reviewer hUJA said they would **consider raising my score**, if answering the questions raised (which we did), some of which involving adding experiments (they mentioned **don't necessarily need the present paper's method to beat this alternative**).
Reviewer 5jc1 asked for two more sets of experiments (included in the response). Reviewer ghPA mentioned some concerns in their comments (we have addressed them by experiments and provided details/clarification in response). Reviewer uP49 would like to see a specific set of experiments in our response (included in the response). Since they asked to include specifically experiments in our response, and we have provided with evidence supporting our method and clarification supporting soundness of our method, we would expect they raise the score accordingly.

Reviewer hUJA think **there's plenty to like about this paper**. They think the paper studies **a problem with lots of practical applications but also elegant theory/methods** and the method is **a neat and clever achievement**. Reviewer 5jc1 said the paper **significantly enhance the scalability of CTHMMs** and **achieves superior results**. Reviewer ghPA said that the idea **appears to be novel**. Reviewer uP49 said the idea **seems sensible and is novel to my knowledge** and **the authors take great care to demonstrate, in the ablation studies, the benefits of different aspects of their methodology in isolation**.  3 of 4 reviewers agree the paper is **clearly written and easy to follow**, and Reviewer uP49 mentioned they **particularly appreciate Figures 1 and 2 which make the proposed approach very clear**. Their main concern is about one of the experiment settings. We train on 80\% of each sequence and test on the full sequences, and reviewers are concerned that the results will be affected by the 80\% training and hide overfiting. We understand their concern so that we add additional results on test part only (calculated from the original experiments), and provided the reason why the settings in our paper - fitting on all available data can be useful for discovering underlying phenomena/dynamics of the sequences (by examining the transition rate/probability matrix and/or emission distributions). Therefore, the performance on all observed data is also valuable historically and practically (e.g. in biology [1]). The added experiment results are still in favour of our method in large complex dataset, but separate validation sets would be helpful for other datasets for performance on unseen parts only.

[1] Krogh, Anders, et al. "Hidden Markov models in computational biology: Applications to protein modeling." Journal of molecular biology 235.5 (1994): 1501-1531.

---

### Meta-Review · Area_Chair_FLHM · 2026-01-01

**Summary:**

All reviewers agree that the method is elegant and well designed. The idea of factorizing the model and using normalizing flows is interesting and clearly presented. However, several important concerns remain.

A main issue raised by all reviewers is the evaluation setup. Models are trained on the first 80% of each sequence and evaluated on full sequences. When evaluation is done only on the held-out 20%, the method often does not outperform the baselines. It is unclear whether the model generalizes well to unseen data or to fully unseen test sequences.

The authors also admit important limitations. The model cannot easily handle asynchronously sampled data. Most experiments are on low-dimensional data (mainly 2D, and simple 3D). It is unclear how the method would scale to general multivariate time series.

Some comparisons are also questionable. The comparison with discrete-time HMMs is not fully fair, and important alternatives such as implicit neural representations and other modern continuous-time models are missing, as pointed out by reviewer hUJA. In addition, some statements in the rebuttal are not precise, for example the claim that “more flows give better results” is unclear about whether it refers to training data or test data.

Overall, while the paper is promising and technically sound, all reviewers (and myself) raised concerns that are not fully addressed. In its current form, the paper is not ready yet.

**Reviewer Concerns:**

**Reviewer Concerns**

**Addressed by the rebuttal**
- Added results on the 20% test part of each sequence.
- Explained how time discretization is done for discrete-time baselines.
- Added comparisons with more continuous-time and probabilistic models.
- Clarified training stability and convergence.
- Explained limits related to higher-dimensional data.

**Still outstanding**
- Generalization is still unclear, since test results are often not better than baselines.
- Evaluation is still based on partly seen sequences, not fully unseen test sequences.
- The model cannot naturally handle asynchronously sampled data.
- Handling of missing data relies on simple imputation.
- Evidence for multivariate time series beyond low-dimensional cases is limited.
- Some rebuttal statements are unclear, such as claims about flows and performance on train vs. test data.

**Reviewer Scores:**

I do not think any reviewer would move its original score, as rebuttal did not satisfactory addressed some of thier comments

---

### Decision · Program_Chairs · 2026-01-26

Reject